# On the Impact of Feature Heterophily on Link Prediction with Graph Neural Networks

**Jiong Zhu**[*†]
University of Michigan
jiongzhu@umich.edu

**Gaotang Li**[*]
University of Illinois
Urbana-Champaign
gaotang3@illinois.edu

**Yao-An Yang**
University of Michigan
ayayang@umich.edu

**Jing Zhu**
University of Michigan
jingzhuu@umich.edu

**Xuehao Cui**
University of Michigan
credus@umich.edu

**Danai Koutra**
University of Michigan
dkoutra@umich.edu

## Abstract

Heterophily, or the tendency of connected nodes in networks to have different class labels or dissimilar features, has been identified as challenging for many Graph Neural Network (GNN) models. While the challenges of applying GNNs for node classification when *class labels* display strong heterophily are well understood, it is unclear how heterophily affects GNN performance in other important graph learning tasks where class labels are not available. In this work, we focus on the link prediction task and systematically analyze the impact of heterophily in *node features* on GNN performance. We first introduce formal definitions of homophilic and heterophilic link prediction tasks, and present a theoretical framework that highlights the different optimizations needed for the respective tasks. We then analyze how different link prediction encoders and decoders adapt to varying levels of feature homophily and introduce designs for improved performance. Based on our definitions, we identify and analyze six real-world benchmarks spanning from homophilic to heterophilic link prediction settings, with graphs containing up to 30M edges. Our empirical analysis on a variety of synthetic and real-world datasets confirms our theoretical insights and highlights the importance of adopting learnable decoders and GNN encoders with ego- and neighbor-embedding separation in message passing for link prediction tasks beyond homophily.

## 1 Introduction

Graph-structured data are powerful and widely used in the real world, representing relationships beyond those in Euclidean data through links. Link prediction, which aims to predict missing edges in a graph, is an important task with applications spanning from recommendation systems [41] to knowledge graphs [38], and social networks [8]. Traditional algorithms for link prediction are heuristic-based and impose strong assumptions on the link generation process. To alleviate the reliance on handcrafted features, recent link prediction approaches are transformed by Graph Neural Networks (GNNs), which can effectively learn node representations in an end-to-end fashion. Vanilla GNN for link prediction (GNN4LP) methods keep the original GNN model for encoding node embeddings, followed by a decoder acting on pairwise node embeddings, *e.g.* dot product [20]. However, these methods are not effective at capturing pairwise structural proximity information [54, 24], *i.e.* neighborhood heuristics such as the number of common neighbors. To further enhance model

---

[*]Equal contribution.  [†]Work done prior to the author's affiliation with Amazon.

38th Conference on Neural Information Processing Systems (NeurIPS 2024).

capabilities, the current *state-of-the-art* GNN4LP approaches augment GNNs by incorporating pairwise structural information [6, 52, 51, 49, 43].

Nevertheless, as the core of today's *SOTA* approaches, GNNs rely on the *message-passing* mechanism which works well when the underlying data exhibit homophily, *i.e.* connected nodes tend to share similar attributes with each other. Such inductive bias has been widely analyzed and has been shown to be an important factor for GNN's superior performance in the task of node classification on homophilic graphs [55, 14, 28, 29]. It has also been widely observed that GNN's performance degrades on heterophilic graphs in node classification tasks, where connected nodes tend to have different labels [1, 58, 60, 34, 17, 27]. In contrast, there are merely works focusing on the problem of heterophily in link prediction tasks: almost all existing definitions of homophily rely on the node class labels [25, 35], which are often not available for the link prediction tasks. Furthermore, prior works on GNN4LP have largely focused on the effects of pairwise structural information to link prediction performance, while there is no dedicated work focusing on the effects of feature heterophily. In light of this, this work aims to characterize the notion of heterophily in the link prediction problem, understand the effects of heterophily and feature similarity in existing models that leverage node features, and explore designs to improve the use of dissimilar features in GNN link prediction. We detail our contributions as follows:

- **Definitions of Non-homophilic Link Prediction**: We introduce formal definitions of homophilic and non-homophilic link prediction tasks: instead of relying on the magnitude of feature similarity, our definitions are based on the separation of feature similarity scores between edges and non-edges, which is justified by a concise theoretical framework that highlights the different optimizations needed for the respective tasks.

- **Designs Empowering GNNs for Non-homophilic Link Prediction**: We identify designs for GNN encoders and link probability decoders that improve performance for non-homophilic link prediction settings and show that (1) decoders with sufficient complexity are required for capturing non-homophilic feature correlations between connected nodes; (2) ego- and neighbor-embedding separation in GNN message passing improves their adaptability to feature similarity variations.

- **Benchmarks for Non-homophilic Link Prediction**: We introduce six real-world datasets spanning from homophilic to heterophilic settings for link prediction, with edge counts ranging from 88K to over 30M, to benchmark the performance of various GNN models. These datasets come from diverse domains and exhibit different levels of feature similarity, providing a robust foundation for evaluating GNN adaptability to non-homophilic conditions in link prediction tasks.

- **Empirical Analysis on Impacts of Feature Heterophily**: We conduct comprehensive empirical analyses using both synthetic and real-world graphs. Through synthetic graphs with controlled feature similarity levels, we analyze how link prediction methods with different designs adapt to varying levels of heterophily. On real-world graphs, we evaluate both overall performance and local behavior across edges with different node degrees and feature similarities, revealing important insights about model adaptability in practice.

## 2   Related Work

**Graph Neural Networks for Link Prediction.** Traditional algorithms for link prediction are primarily *heuristic-based*, which have strong assumptions on the link generation process. These approaches compute the similarity scores between two nodes based on certain structures or properties [2, 3, 5, 51]. Later, various *representation-based* algorithms for link prediction were proposed, which aim at learning low-dimensional node embeddings that are used to predict the likelihood of link existence between certain node pairs and usually involve the use of GNNs [21, 15, 42]. Compared with the *heuristic-based* algorithms, *representation-based* algorithms do not require strong assumptions and perform learning over the graph structure and node features in a unified way. A representative method is the Variational Graph Autoencoder [20], which uses GCN as the encoder for learning node representations and inner product as the decoder for pairwise link existence predictions. More recently, the state-of-the-art methods for link prediction are built on top of the *representation-based* algorithms and augment them with additional *pairwise* information. For instance, *subgraph-based* approaches perform link prediction between two nodes by first extracting their enclosing subgraphs and subsequently applying the standard *representation-based* algorithms on the extracted subgraphs [52, 6, 62]. Concurrently, other works have been proposed to augment GNN learning with common neighbor

information [43, 49, 43]. Despite the tremendous success, prior link prediction works mainly assume homophily, where node pairs with similar features or neighbors are more likely to link together. In contrast, this work considers a more general setting where there may be a spectrum of low-to-high homophily in the underlying data, characterizes a notion of heterophily in link prediction and explores how popular approaches perform under this characterization. Related work also examines the influence of data for link prediction from a joint perspective of graph structure and feature proximity [30, 22]. While these works provide valuable insights into the interplay between structural and feature information, our work specifically focuses on characterizing and analyzing the impact of node feature through the lens of heterophily, offering complementary insights into how varying degrees of feature similarity affect link prediction performance.

**Graph Neural Networks Addressing Heterophily.** There is a rich literature on graph neural networks addressing heterophily, but most of them tackle the node classification task [1, 58, 60, 34, 17, 27]. Very few works focus on the problem of link prediction under heterophily. Among them, Zhou et al. [56] propose to disentangle the node representations from latent dissimilar factors, and Di Francesco et al. [9] extend the physics-inspired GRAFF [10], originally designed for handling heterophilic node classification tasks, to heterophilic link prediction tasks. These works use the same *class-dependent* homophily measure as typically used in node classification, while our work emphasizes the influence of features in link prediction and more systematically benchmarks model performance against different homophily levels.

# 3 Notation and Preliminaries

Let $\mathcal{G} = (\mathcal{V}, \mathcal{E})$ be an undirected and unweighted homogeneous graph with node set $\mathcal{V}$ and edge set $\mathcal{E}$. We denote the 1-hop (immediate) neighborhood centered around $v$ as $N(v)$ ($\mathcal{G}$ may have self-loops), and the corresponding neighborhood that does *not* include the ego (node $v$) as $\bar{N}(v)$. We represent the graph by its adjacency matrix $\mathbf{A} \in \{0, 1\}^{n \times n}$ and its node feature set as $\mathcal{X}$ with matrix form $\mathbf{X} \in \mathbb{R}^{n \times F}$, where the vector $\mathbf{x}_v$ corresponds to the *ego-feature* of node $v$, and $\{\mathbf{x}_u : u \in \bar{N}(v)\}$ to its *neighbor-features*. We further represent the degree of a node $v$ by $d_v$, which denotes the number of neighbors in its immediate neighborhood $\bar{N}(v)$.

**Graph Neural Networks for Link Prediction**. Following [23, 53], we define the task of link prediction to be estimating the likelihood of reconstructing the actual adjacency matrix. Formally,

$$\hat{y}_{i,j} = \hat{\mathbf{A}}_{i,j} = p(i, j | \mathcal{G}, \mathbf{X}), \tag{1}$$

where $\hat{y}_{i,j}$ or $\hat{\mathbf{A}}_{i,j}$ is the predicted link probability between nodes $(i, j)$ and was traditionally calculated by heuristics-based algorithms. For $(i, j)$ in the training set, we set the ground truth probability $y_{i,j} = \mathbf{A}_{i,j}$. Existing GNN4LP approaches typically use a GNN-based method for encoding node representations (denoted by ENC) and some decoder function (denoted by DEC) between node embedding pairs:

$$\hat{y}_{i,j} = \hat{\mathbf{A}}_{i,j} = \text{DEC}(z_i, z_j), \text{ where } z_i = \text{ENC}(i, \mathcal{G}, \mathbf{X}), z_j = \text{ENC}(j, \mathcal{G}, \mathbf{X}). \tag{2}$$

The original graph autoencoder approach [20] uses a two-layer GCN [21] as the encoder and a dot product as the decoder. There are many different choices of encoders that do not need to strictly follow the original GNN architecture. For the decoder, while dot product remains a popular simple choice [20, 33], more expressive alternatives include concatenation followed by an MLP.

**Graph Feature Similarity**. We measure the "graph feature similarity" through averaging the *mean-centered* feature similarity from connected node pairs.

**Definition 1 (Node Feature Similarity)** *For a node pair $(u, v)$ with $\mathbf{x}_u$ and $\mathbf{x}_v$ as the node features and a similarity function $\phi \colon (\cdot, \cdot) \mapsto \mathbb{R}$, we define the node feature similarity as $k(u, v) = \phi(\mathbf{x}_u, \mathbf{x}_v)$.*

In this work, we set $\phi(\mathbf{x}_u, \mathbf{x}_v) = \frac{\bar{\mathbf{x}}_u \cdot \bar{\mathbf{x}}_v}{\|\bar{\mathbf{x}}_u\| \|\bar{\mathbf{x}}_v\|}$ to be the *mean-centered* cosine similarity of node features, where we denote the mean feature vector of all nodes in the graph as $\bar{\mathbf{x}} = \frac{1}{|\mathcal{V}|} \sum_{v \in \mathcal{V}} \mathbf{x}_v$, and the mean-centered node feature for node $v$ as $\bar{\mathbf{x}}_v = \mathbf{x}_v - \bar{\mathbf{x}}$. Empirically, we find that the mean-centering operation is crucial for accurate characterization of the pairwise feature similarity and its impact on link prediction performance. We further define the graph feature similarity as follows.

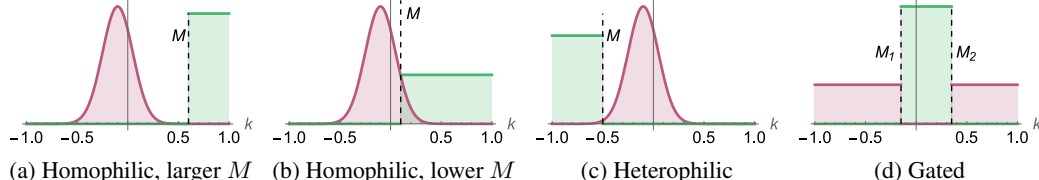

Figure 1: Categorizing link prediction tasks based on the distribution of feature similarity scores of positive node pairs (i.e., edges – colored in green) and negative node pairs (non-edges – colored in red): two distributions whose density is visualized in the plots are (approximately) separated by the threshold(s) $M$. Homophilic and heterophilic link prediction differs in whether the positive similarity scores fall into the larger or smaller side of the threshold $M$, while the magnitude of $M$ indicates the variance of positive similarity. Gated link prediction is a more complex case where the distribution of positive and negative similarity scores cannot be separated by a single threshold.

**Definition 2 (Graph Feature Similarity)** *We measure the graph feature similarity $K$ through averaging the feature similarity of all its connected nodes pairs:* $K = \sum_{(u,v) \in \mathcal{E}} \frac{k(u,v)}{|\mathcal{E}|}$.

Unlike the homophily measures defined on node class labels which are non-negative [25], the feature similarity $k(u, v) \in [-1, 1]$ can additionally be *negative*, indicating negative correlations. We refer to the graph as *positively correlated* if $K > 0$ and *negatively correlated* if $K < 0$.

## 4 Homophilic & Heterophilic Link Prediction

While existing works on node classification usually define homophilic or heterophilic graphs by whether the majority of connected nodes share the same class labels [25, 35], the class information is *not* available for link prediction. Instead, we argue that the homophilic and heterophilic link prediction tasks should be defined based on how the distributions of feature similarity scores between connected and unconnected nodes are separated, as these definitions capture the fundamental differences on how link prediction scores are correlated with feature similarity scores. Within the category of homophilic or heterophilic tasks, we further show that the variation of the positive feature similarity scores affects the rate of change for the link prediction scores. We present definitions with intuitive examples in §4.1 and theoretical analysis in §4.2.

### 4.1 Categorizing Link Prediction on Distributions of Feature Similarity

We begin our discussion by considering the distributions of feature similarity scores for a random positive (edge) and negative (non-edge) node pair in the graph: consider the set of feature similarity scores for positive (edge) node pairs in the graph as $\mathcal{K}_{pos}$, and negative (non-edge) node pairs as $\mathcal{K}_{neg}$. We first formalize different categories of link prediction tasks, which are defined by how the distributions of $\mathcal{K}_{pos}$ and $\mathcal{K}_{neg}$ are (approximately) separated:

**Definition 3 (Homophilic Link Prediction)** *The task is* homophilic *if $M \in \mathbb{R}$ exists such that for most samples $\tilde{\mathcal{K}}_{pos} \subset \mathcal{K}_{pos}$ and $\tilde{\mathcal{K}}_{neg} \subset \mathcal{K}_{neg}$ it satisfies $\sup(\tilde{\mathcal{K}}_{neg}) < M \leq \inf(\tilde{\mathcal{K}}_{pos})$.*

Prior works have mostly focused on the homophilic category for the link prediction problem while overlooking other possibilities. For other cases where the homophilic conditions are not satisfied, we refer to them generally as *non-homophilic* link prediction problems. In the definition below, we formalize an easy type of non-homophilic link prediction problem:

**Definition 4 (Heterophilic Link Prediction)** *The task is* heterophilic *if $M \in \mathbb{R}$ exists such that for most samples $\tilde{\mathcal{K}}_{pos} \subset \mathcal{K}_{pos}$ and $\tilde{\mathcal{K}}_{neg} \subset \mathcal{K}_{neg}$ it satisfies $\sup(\tilde{\mathcal{K}}_{pos}) \leq M < \inf(\tilde{\mathcal{K}}_{neg})$.*

We give intuitive examples of homophilic and heterophilic link prediction tasks in Fig. 1: the key difference between homophilic and heterophilic tasks is whether $\mathcal{K}_{pos}$ is predominantly distributed above threshold $M$ while $\mathcal{K}_{neg}$ is below $M$ (homophilic), or vice versa (heterophilic), as shown in Fig. 1a vs. 1c. The categorizations of homophilic/heterophilic link prediction tasks should not be confused with the magnitude of $M$ that indicates the variance of positive similarity scores: while its

magnitude does not determine the type of the link prediction problem, our analysis in the next section does show that it affects the rate of change for the link prediction scores.

However, beyond the heterophilic setting defined above, there are other non-homophilic settings with even more complexity, where the distribution of $\mathcal{K}_{pos}$ and $\mathcal{K}_{neg}$ cannot be separated by a single threshold $M$. Most of these cases are too complex to be formalized and studied theoretically, but we formalize one of them (Fig. 1d) below and later report empirical results (§ 6).

**Definition 5 (Gated Link Prediction)** *The task is* gated *if it is neither homophilic nor heterophilic, but $M_1, M_2 \in \mathbb{R}$ exist such that $M_2 \geq \sup(\tilde{\mathcal{K}}_{pos}) \geq \inf(\tilde{\mathcal{K}}_{pos}) \geq M_1$.*

### 4.2 Differences between Homophilic & Heterophilic Link Prediction

In §4.1, we have situated our discussions of link prediction tasks based on how the positive and negative samples are separated in the feature similarity score space. In this section, we reveal on a stylized learning setup the fundamental differences in optimizations for homophilic and heterophilic link prediction tasks.

**Theoretical Assumptions**. *Assume a training graph with node $v \in \mathcal{V}$ whose feature vectors are 2-dimensional unit vectors and can be represented as $\mathbf{x}_v = (\cos\theta_v, \sin\theta_v)$. We consider a DistMult decoder for predicting the link score for candidate node pair $u, v \in \mathcal{V}$ with feature vectors $\mathbf{x}_u, \mathbf{x}_v$. Specifically, the link score is calculated as $\hat{y}_{u,v} = (\mathbf{x}_u \otimes \mathbf{x}_v)^\mathsf{T}\mathbf{w} + b$, where $\mathbf{w}$ and $b$ are learnable parameters for the decoder, with training loss function $\mathcal{L} = y \cdot \mathrm{ReLU}(-\hat{y}) + (1-y) \cdot \mathrm{ReLU}(\hat{y})$ such that $\hat{y} \geq 0$ for all edges (positive samples) and $\hat{y} < 0$ otherwise. Furthermore, we assume that the $\mathcal{K}_{pos}$ and $\mathcal{K}_{neg}$ are* ideally separable *by a threshold $M \in [0, 1]$ in the feature similarity score space such that the homophilic or heterophilic conditions hold for* all *samples.*

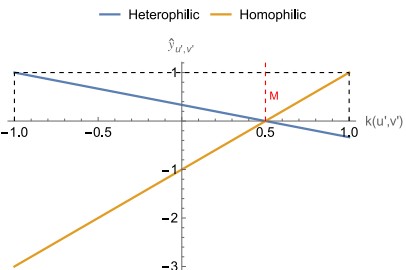

Figure 2: Link prediction scores $\hat{y}_{u'v'}$ for decoders optimized under homophilic (yellow) and heterophilic (blue) setups in Thm. 1 (for $M = 0.5$).

With the above assumptions, we now show that (1) the predicted link score and feature similarity scores are positively correlated for homophilic tasks, while negatively correlated for heterophilic tasks; (2) the change rate for the predicted link probability with respect to the feature similarity is determined by the magnitude of the threshold $M$ that separates the positive and negative samples.

**Theorem 1** *Following the above assumptions, consider two DistMult decoders that are fully optimized for homophilic and heterophilic link prediction problems respectively. Given an arbitrary node pair $(u', v')$ with node features $\mathbf{x}_{u'} = (\cos\theta_{u'}, \sin\theta_{u'})$ and $\mathbf{x}_{v'} = (\cos\theta_{v'}, \sin\theta_{v'})$ and pairwise feature similarity $k(u', v')$, the following holds for the predicted link probability $\hat{y}_{u'v'}$:*

- *For the homophilic problem where $\sup(\mathcal{K}_{neg}) < \inf(\mathcal{K}_{pos}) = M \leq 1$, when bounding $\hat{y}_{u,v} = 1$ if $k(u,v) = 1$ during training, $\hat{y}_{u'v'}$ increases with $k(u', v')$ at a linear rate of $\frac{1}{(1-M)}$;*

- *For the heterophilic problem where $-1 \leq \sup(\mathcal{K}_{pos}) = M < \inf(\mathcal{K}_{neg})$, when bounding $\hat{y}_{u,v} = 1$ if $k(u,v) = -1$ during training, $\hat{y}_{u'v'}$ decreases with $k(u', v')$ at a linear rate of $\frac{1}{(1-M)}$.*

We give the proof in App. §B.1 and visualize in Fig. 2 how the predicted link score $\hat{y}_{u'v'}$ changes under the homophilic and heterophilic settings. Though the above results are derived under simplified assumptions, it clearly highlights the different optimizations needed for homophilic and heterophilic link prediction tasks that have not been studied in prior literature. In §6, we observe that these differences go beyond our theoretical assumptions and affect the performance of all GNN encoders and link prediction decoders on datasets with higher complexity, which warrant our study of effective encoder and decoder choices for non-homophilic link prediction in the next section.

## 5 Encoder & Decoder Choices for Link Prediction Beyond Homophily

In §4, we gave formal definitions of homophilic and heterophilic link prediction tasks and highlight their differences in model optimizations. As non-homophilic settings are largely overlooked in prior

literature, we aim to verify whether existing GNN message passing designs for node features remain effective beyond homophily. We follow the encoder-decoder perspective in [16] and discuss designs for both GNN encoder and link prediction decoder that adapt to non-homophilic settings.

## 5.1 Decoder Choice for Heterophilic & Gated Link Prediction

For *homogeneous* graphs that only have one edge type (as opposite to heterogeneous or knowledge graphs), popular decoder choices for deriving link probability from node representations are either a simple dot product (DOT) operation or more complex multi-layer perceptron (MLP). While the MLP decoder has a stronger representation power due to its non-linearity, the inner product decoder is more preferred in large-scale applications due to its fast inference speed: it is well established that maximum inner product search (MIPS) can be approximated with sublinear complexity using packages such as Faiss [12]. A prior work [44] has benchmarked the performance of different link prediction decoders on several OGB datasets [18] and proposed a sublinear approximation of MLP decoder during inference time. However, no study has been conducted on the performance of decoders for non-homophilic link prediction tasks across the negative to positive similarity spectrum.

Our takeaways for effective decoder choices for non-homophilic link prediction tasks are as follows: (1) for non-homophilic (e.g., gated) tasks, only non-linear decoders such as MLP are suitable; (2) for heterophilic tasks, a linear decoder with learnable weights (e.g., DistMult [47]) can be used in lieu of MLP to achieve better scalability while maintaining comparable performance; (3) dot product decoder is only suitable for homophilic link prediction tasks.

Theoretically, we formalize our first takeaway with the below theorem, which shows the limitations of using linear decoders (such as DOT and DistMult) in non-homophilic link prediction tasks:

**Theorem 2** *No parameter exists for a single linear decoder that perfectly separates link probability for edges and non-edges for gated link prediction.*

We give the proof in Appendix §B.2. For linear models, while both DistMult and DOT product decoders share the same time complexity during inference, we observe empirically in §6 that DistMult outperforms DOT decoder by up to 55% on non-homophilic link prediction tasks. Intuitively, the learnable weights in DistMult decoder allow the model to capture the negative correlation between connected node features and improve its effectiveness for heterophilic tasks.

## 5.2 Improving GNN Representation Power with Heterophily-adjusted Designs

We now consider the impact of GNN architectures on non-heterophilic link prediction performance. In particular, we examine whether the effective designs for node classification beyond *class* homophily can be transferred to link prediction tasks beyond *feature* homophily. A design that significantly improves classification performance under low class homophily is the separation of ego- and neighbor-embeddings in GNN message passing, which has consistently shown to improve classification performance across multiple studies [60, 36]. As real-world graphs usually follow a power-law degree distribution and exhibit large variation in node degrees, prior work has used the robustness of GNN models to degree shift as a proxy to measure the generalization ability of GNN models for heterophilic node classification [60]. We follow a similar approach in the theorem below and show that a GNN model that embeds ego- and neighbor-features together is less capable of generalizing under heterophilic settings than a graph-agnostic model. We give the proof in Appendix §B.3.

**Theorem 3** *Consider the same DistMult decoder and loss function $\mathcal{L}$ as the assumptions in §4.2, but trained on a heterophilic graph where (1) feature vectors for all nodes can be either $\mathbf{x}_1 = (\cos\theta_1, \sin\theta_1)$ or $\mathbf{x}_2 = (\cos\theta_2, \sin\theta_2)$, and (2) nodes $u$ and $v$ are connected if and only if $\mathbf{x}_u \neq \mathbf{x}_v$. Assume two DistMult decoders are trained, one baseline with node features $\mathbf{x}_u$, and the other with GNN representations $\mathbf{r}_u$ instead of node features $\mathbf{x}_u$, where $\mathbf{r}_u$ is obtained with a linear GNN model $\mathbf{r}_u = \frac{1}{|\bar{N}(u)|+1}\mathbf{x}_u + \frac{1}{|\bar{N}(u)|+1}\sum_{l\in\bar{N}(u)}\mathbf{x}_l$ that considers self-loops in its message passing process. We further assume a degree shift between training and test sets, where all training nodes have degree $d$ while the test nodes have degree $d'$. Then for any $d' > 0$ when $d = 0$, or $1 \leq d' < d$ when $d \geq 2$, the DistMult decoder optimized on GNN representations $\mathbf{r}_u$ reduce the separation distance between edges and non-edges for the test nodes compared to the baseline optimized with node features $\mathbf{x}_u$.*

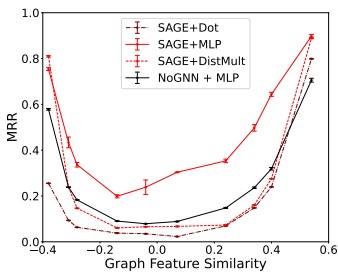 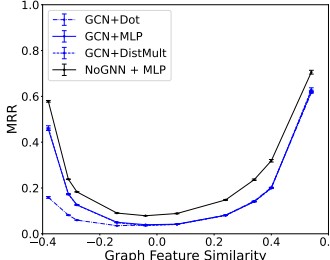 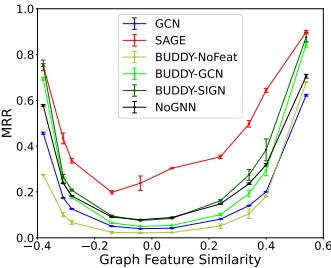

(a) Comparison of decoder choices with SAGE encoder.

(b) Comparison of decoder choices with GCN encoder.

(c) Comparison of different encoder choices with MLP decoder.

Figure 3: Comparing link prediction methods on synthetic graphs with varying levels of feature similarity: (a) and (b) focus on decoders, while (c) focuses on encoders. We include MLP decoder without GNN as a graph agnostic baseline in all plots. Numerical results are reported in Table 2.

# 6 Empirical Analysis

We aim to understand through empirical analysis (1) what are the performance trends of link prediction methods under different link prediction tasks in the full spectrum of negative to positive feature similarity, and (2) how different encoder and decoder designs adapt to non-homophilic link prediction tasks, including variations of feature similarity and node degrees within the same graph. We first introduce the link prediction methods that we consider in our experiments, and then present the results on synthetic and real-world datasets. More details about setups and results are available in App. A.[3]

**Link Prediction Methods**. As in the previous sections, we follow an encoder-decoder framework and consider different combinations of both components. For *decoders*, we consider the options mentioned in §5.1: (1) **Dot Product (DOT)**, (2) **Multi-Layer Perceptron (MLP)**, and (3) **DistMult** [47].

For *encoders*, we consider two GNN methods to study how separating ego- and neighbor-embeddings affects link prediction performance: (1) **GraphSAGE** [15] which separates ego- and neighbor-embeddings during message passing, and (2) **Graph Convolutional Network (GCN)** [21], which does not make this separation. We couple these encoders with the decoders above to form six GNN4LP models.

Furthermore, to understand how message passing designs affect performance for GNN4LP models that leverage both node features and pairwise structural information, we consider BUDDY [6], a state-of-the-art method reported in a recent benchmark [23]. BUDDY augments GNN encoders with subgraph sketching to capture structural information. Specifically, we consider three variants: (3) **BUDDY-GCN**, which uses GCN for feature aggregation, (4) **BUDDY-SIGN**, which uses SIGN aggregation [38] to separate ego- and neighbor-embeddings during message passing, and (5) **NoFeat**, a structure-only baseline that excludes node features and relies solely on structural information for link prediction. All BUDDY variants use an MLP decoder as part of their architecture.

Finally, we also consider these link prediction heuristics tested in [6]: **Common Neigbhors (CN)** [32], **Resource Allocation (RA)** [57], **Adamic-Adar (AA)** [2], and **Personalized PageRank (PPR)** [19].

## 6.1 Experiments on Synthetic Graphs

We generate synthetic graphs that resemble different types of link prediction tasks (i.e., homophilic, heterophilic, and gated) by varying the feature similarity between connected nodes. These graphs provide controlled environments that allow us to focus on the effects of feature similarity on link prediction performance without mingling them with other data factors such as structural proximity. We give the details of the synthetic graph generation process and the experiment setup in App. §A.

**Performance Trend Across the Full Similarity Specturm**. We visualize the performance trends per method in Fig. 3 and present the numerical results in Table 2. We observe that the performance of all feature-consuming methods is significantly affected by the level of feature similarity: most methods reach their best performance at the positive extreme (homophilic tasks) and the second best at the

---

[3] Our experiment code is available at `https://github.com/GemsLab/HeteLinkPred`.

negative extreme (heterophilic tasks); between the two extremes (gated tasks), the performance of all methods drops significantly as the feature similarity score approaches 0, which creates a U-shaped performance trend across the feature similarity spectrum. It is worth noting that graph-agnostic MLP decoder without GNN (NoGNN) also exhibits the U-shaped performance trend, which suggests the challenges of leveraging node features effectively in these settings for non-GNN methods as well. Intuitively, as the similarity scores for a random pair of nodes follow a normal distribution centered around 0, the performance drop when average feature similarity scores approach 0 can be contributed to the reduced distinguishability between the similarity scores of edges and non-edges in the graph. For feature agnostic heuristics such as Common Neighbors and Personalized PageRank, they show mostly unproductive performance except at the positive extreme. This suggests that graphs formed by positive feature correlations are also likely to show strong structural proximity, which is beneficial for the heuristic-based methods for the homophilic link prediction task.

**Decoder Choices: MLP and DistMult over DOT**. We further validate our main points in §5 regarding how different decoder choices adapt to non-homophilic link prediction settings. In Fig. 3a-b, we compare the performance of different decoders with fixed SAGE and GCN encoders, respectively. With both SAGE and GCN encoders, we observe that DOT decoder performs the worst among all choices across all feature similarity levels: it is outperformed by MLP decoder with a margin of 50% in the negative extreme and 10% in the positive extreme under SAGE encoder. DistMult decoder performs significantly better than the DOT decoder, especially in the region of negative feature similarity: at the negative extreme, DistMult decoder outperforms DOT decoder by 55% and even outperforms MLP decoder by 5.6%. Empirically, we observe that the optimization process of DistMult is more stable than MLP when using a SAGE encoder in the negative similarity region, allowing it to reach optimal performance without suffering from instabilities at the negative extreme. However, the performance of DistMult coupled with SAGE decoder is significantly lower than MLP with up to 37% gap between the negative and positive extremes, where the link prediction tasks are gated instead of being homophilic or heterophilic. This validates our theoretical analysis in §5.1 that the linear decoders like DOT and DistMult are not suitable for the settings where non-linear separation between similarity scores of edges and non-edges are required. With GCN encoder, the performance of DistMult decoder is mostly on par with MLP decoder across the spectrum, which shows that the performance bottleneck is on the encoder side rather than the decoder side. Overall, MLP decoder is the most robust choice across different feature similarity levels and link prediction tasks, yielding the best performance in all but one cases when coupled with SAGE encoder, with DistMult being a more scalable alternative for homophilic and heterophilic link prediction tasks.

**Encoder Choices: Importance of Ego- and Neighbor-Embedding Separation.** In Fig. 3c, we compare the performance of different encoder choices with fixed MLP decoder, which is the best-performing decoder option for nearly all cases. In addition to GCN and SAGE encoders that rely only on node features, we also include variants of BUDDY [6] that leverage structural proximity and (optionally) node features. Comparing between SAGE and GCN encoders, we observe that SAGE consistently outperforms GCN across all feature similarity levels by up to 44%. For BUDDY variants with SIGN and GCN feature encoders, we also observe consistently better performance with SIGN encoder across all feature similarity levels with up to 9.0% gain. Both comparisons suggest the importance of adopting ego- and neighbor-embedding separation in GNN encoder design for link prediction: as discussed in §5.2, this design allows GNN encoders to learn representations that are more robust to variations of node degrees and feature similarity levels in the graph instead of overfitting to specific degrees or similarity scores, which we also observe in the real-world datasets.

**Importance of Node Features vs. Structural Proximity**. Here we compare two approaches in Fig. 3c: (1) SAGE+MLP, which combines node features with implicit graph structural information captured through GNN message passing, and (2) BUDDY-SIGN, which additionally incorporates explicit structural proximity (e.g., number of shared neighbors) through subgraph sketching. We find that BUDDY-SIGN's performance is consistently lower than SAGE+MLP across all feature similarity levels, with the gap reaching up to 26% between the negative and positive extremes. This suggests that when graph connections are predominantly driven by feature similarity (as in our synthetic graphs), the additional structural information from subgraph sketching may not provide added benefit beyond the structural information already captured by GNN message passing. While real-world graphs are typically influenced by both feature similarity and structural proximity, these results emphasize the importance of carefully balancing these two information sources in link prediction models, particularly for non-homophilic settings where structural proximity alone may be less informative.

Table 1: Results on real-world graphs. "*" denotes results quoted from [6].

| Dataset | ogbl-collab | ogbl-citat2 | e-comm | facebook | PPI | amzn-comp |
|---|---|---|---|---|---|---|
| #Nodes | 235,868 | 2,927,963 | 346,439 | 4,039 | 56,944 | 13,752 |
| #Edges | 2,358,104 | 30,387,995 | 682,340 | 88,234 | 1,612,348 | 491,722 |
| Feat. Sim | $0.70_{\pm 0.23}$ | $0.40_{\pm 0.22}$ | $0.18_{\pm 0.63}$ | $0.11_{\pm 0.23}$ | $0.11_{\pm 0.46}$ | $0.07_{\pm 0.35}$ |
| Metrics | Hits@50 | MRR | MRR | MRR | MRR | MRR |
| HEURISTICS | | | | | | |
| CN | 56.44* | 51.47* | 19.96 | 53.83 | 62.77 | 55.05 |
| AA | 64.35* | 51.89* | 19.96 | 54.90 | 64.66 | 57.94 |
| RA | 64.00* | 51.98* | 19.96 | 55.50 | 64.26 | 58.03 |
| DOT DECODER | | | | | | |
| GCN | $10.64_{\pm 0.42}$ | $40.38_{\pm 1.52}$ | $33.83_{\pm 1.34}$ | $39.95_{\pm 0.14}$ | $13.35_{\pm 0.43}$ | $24.67_{\pm 0.60}$ |
| SAGE | $19.71_{\pm 0.52}$ | $71.39_{\pm 0.28}$ | $52.30_{\pm 5.11}$ | $49.59_{\pm 0.05}$ | $18.94_{\pm 0.17}$ | $42.25_{\pm 0.54}$ |
| DISTMULT DECODER | | | | | | |
| GCN | $25.62_{\pm 0.82}$ | $62.31_{\pm 0.68}$ | $53.09_{\pm 2.37}$ | $45.63_{\pm 0.19}$ | $14.31_{\pm 0.29}$ | $31.59_{\pm 1.04}$ |
| SAGE | $43.50_{\pm 1.13}$ | $82.26_{\pm 0.02}$ | $50.15_{\pm 6.56}$ | $50.64_{\pm 0.19}$ | $22.26_{\pm 0.35}$ | $58.42_{\pm 0.12}$ |
| MLP DECODER | | | | | | |
| NoGNN | $6.07_{\pm 0.18}$ | $27.64_{\pm 0.21}$ | $24.65_{\pm 0.21}$ | $16.57_{\pm 0.08}$ | $1.96_{\pm 0.01}$ | $21.15_{\pm 0.23}$ |
| GCN | $30.17_{\pm 2.90}$ | $73.57_{\pm 0.35}$ | $54.82_{\pm 3.42}$ | $49.02_{\pm 0.23}$ | $17.67_{\pm 2.28}$ | $52.90_{\pm 0.86}$ |
| SAGE | $48.64_{\pm 0.39}$ | $83.67_{\pm 0.07}$ | $54.60_{\pm 0.09}$ | $50.12_{\pm 0.05}$ | $35.30_{\pm 0.55}$ | $58.38_{\pm 0.07}$ |
| BUDDY WITH MLP DECODER | | | | | | |
| NoFeat | $66.06_{\pm 0.22}$ | $83.36_{\pm 0.14}$ | $6.68_{\pm 0.00}$ | $42.06_{\pm 7.87}$ | $66.42_{\pm 0.03}$ | $53.53_{\pm 0.02}$ |
| GCN | $66.21_{\pm 0.33}$ | $87.05_{\pm 0.04}$ | $13.13_{\pm 1.91}$ | $52.99_{\pm 0.13}$ | $65.12_{\pm 0.49}$ | $60.99_{\pm 0.33}$ |
| SIGN | $66.64_{\pm 0.64}$ | $87.53_{\pm 0.12}$ | $10.95_{\pm 3.68}$ | $51.81_{\pm 0.13}$ | $53.22_{\pm 0.21}$ | $60.05_{\pm 0.42}$ |

## 6.2 Experiments on Real-world Graphs

Next we compare performance of different link prediction methods on real-world graphs of varying sizes and feature similarities. Unlike our synthetic graphs where the feature similarity of connected nodes is controlled, real-world graphs tend to have a significantly larger variation in feature similarity across edges. Our analysis aims to not only compare the overall performance of the methods, but also to understand their local performance discrepancies across feature similarity variations within a graph, which is typically overlooked in the literature.

**Experiment Setups**. Based on our definitions (§4.1), we employ 6 real-world datasets spanning from homophilic to non-homophilic link prediction tasks: ogbl-collab [18], ogbl-citation2 [18], e-comm [61], facebook [39], PPI [63], and amazon-computers [40]. The details and experiment setup are in App. §A. We report dataset statistics in Table 1 and show feature similarity distributions of edges vs. random node pairs in Fig. 6 (App. A): while ogbl-collab and ogbl-citation2 are approximately homophilic, e-comm, facebook, PPI and amazon-computers are non-homophilic. We have also identified additional real-world datasets exhibiting feature heterophily in App. §C.

Motivated by prior findings about the interplay of node degree and heterophily [46, 59, 26], we analyze how different encoder and decoder choices perform across edges with varying graph properties. Specifically, we group edges into buckets based on two key properties: (1) the degrees of their connected nodes and (2) their feature similarity scores. For each bucket, we compute the average link prediction performance of different methods, with detailed methodology in App. §A. To understand how encoders and decoders adapt to these property variations, we fix one component (encoder or decoder) while varying the other, and visualize the resulting performance differences across buckets. The main results are shown in Fig. 5, with additional analyses in Fig. 7-10.

**Significance of Decoder Choices.** We first compare the performance of different decoder choices while fixing the encoder. As we observed on the synthetic datasets, for both GCN and SAGE encoders, MLP has the highest overall performance, followed by DistMult and DOT. In terms of the robustness to local variations of node degrees and feature similarity scores, we observe in Fig. 5c that despite MLP only outperforms DOT by 2.3% on the full test split, it outperforms DOT on the majority of the feature similarity range by up to 7.4% as DOT overfits to the lowest feature similarity bucket; we also

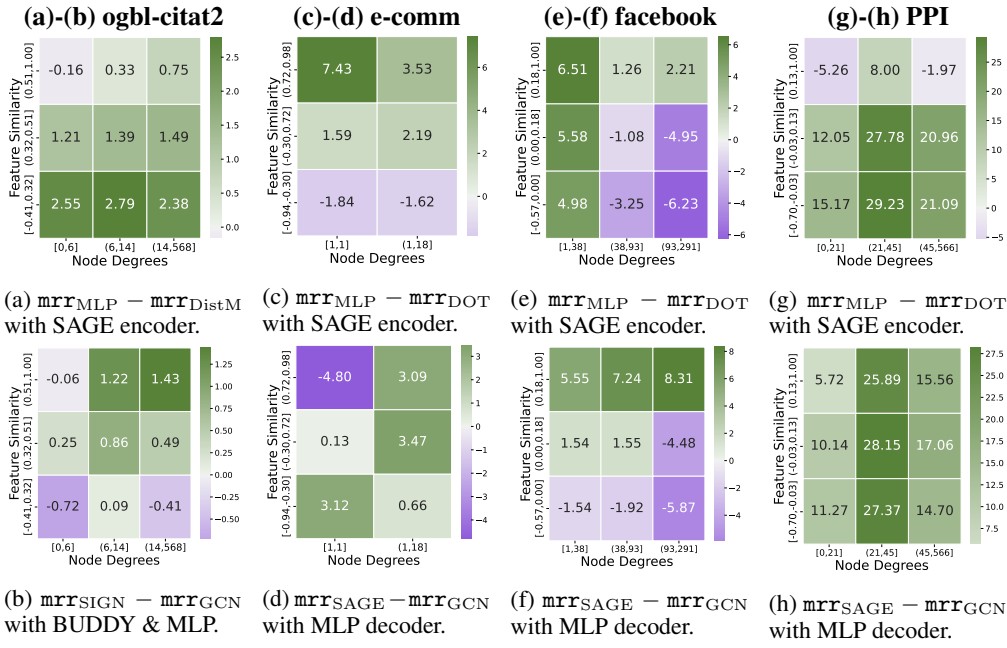

**(a)-(b) ogbl-citat2**  **(c)-(d) e-comm**  **(e)-(f) facebook**  **(g)-(h) PPI**

(a) $\mathrm{mrr_{MLP}} - \mathrm{mrr_{DistM}}$ with SAGE encoder.

(c) $\mathrm{mrr_{MLP}} - \mathrm{mrr_{DOT}}$ with SAGE encoder.

(e) $\mathrm{mrr_{MLP}} - \mathrm{mrr_{DOT}}$ with SAGE encoder.

(g) $\mathrm{mrr_{MLP}} - \mathrm{mrr_{DOT}}$ with SAGE encoder.

(b) $\mathrm{mrr_{SIGN}} - \mathrm{mrr_{GCN}}$ with BUDDY & MLP.

(d) $\mathrm{mrr_{SAGE}} - \mathrm{mrr_{GCN}}$ with MLP decoder.

(f) $\mathrm{mrr_{SAGE}} - \mathrm{mrr_{GCN}}$ with MLP decoder.

(h) $\mathrm{mrr_{SAGE}} - \mathrm{mrr_{GCN}}$ with MLP decoder.

Figure 5: Pairwise comparison of encoder or decoder choices on test edges grouped by node degrees (x-axis) and feature similarity (y-axis): Green denotes MRR increases and purple denotes decreases. More plots in Fig. 7-10.

observe similar overfitting of DistMult to the highest feature similarity bucket in Fig. 5a, despite it being significantly more robust than DOT. Overall, MLP is the most robust choice for link prediction tasks on real-world graphs with varying feature similarity levels.

**Significance of Encoder Choices.** We finally compare different encoder choices while fixing the decoder. Matching our observations on synthetic datasets, we find that SAGE encoder outperforms the GCN encoder by significant margin under most datasets and decoder choices. On e-comm dataset, the performance of GCN on the full test split is on-par with SAGE under MLP decoder, but we observe in Fig. 5d that GCN largely overfits to the edges in the high feature similarity and low degree bucket, with SAGE outperforming GCN by up to 3.5% in the remaining buckets. The similar overfitting is also observed on SIGN vs. GCN with BUDDY and MLP decoder on ogbl-citation2 (Fig. 5b). These observations show that the separation of ego- and neighbor-embeddings in the SAGE and SIGN encoders help GNNs to better adapt to local variations in feature similarity and node degrees in the real-world graphs.

## 7 Conclusion

We characterized non-homophilic link prediction through the distributions of feature similarities between linked and unlinked nodes, and proposed a theoretical framework highlighting the optimizations needed for different tasks. Our analysis revealed how link prediction encoders and decoders adapt to varying feature homophily levels, identifying key designs—learnable decoders (e.g., MLP or DistMult) with GNN encoders that separate ego- and neighbor-embeddings—for improved link prediction performance beyond homophily. Experiments on synthetic and real-world datasets demonstrated the effectiveness of these designs across the feature similarity spectrum.

In summary, our work advances the understanding of heterophilic link prediction, and lays the groundwork for future research. First, we believe that there is a need for introducing more feature-heterophilic benchmark datasets for link prediction. Expanding the diversity of available datasets would enhance the generality and practical significance of studies in this area, allowing for more robust validation of methodologies and mitigating potential biases from the scarcity of strongly heterophilic benchmarks. Second, additional in-depth theoretical frameworks could provide a more comprehensive understanding of the complexities inherent in real-world networks. This includes exploring the interplay between feature similarity and structural similarity and how these relationships influence the performance of different link prediction methods.

## Acknowledgments and Disclosure of Funding

We thank Yuhang Zhou for sharing the experiment code for SpotTarget [61], which served as the foundation for our experiment implementation. We also thank the reviewers for their constructive feedback. This material is based upon work supported by the National Science Foundation under IIS 2212143 and CAREER Grant No. IIS 1845491. We gratefully acknowledge the support of NVIDIA Corporation with the donation of the Quadro P6000 GPU used for this research. Any opinions, findings, and conclusions or recommendations expressed in this material are those of the author(s) and do not necessarily reflect the views of the National Science Foundation or other funding parties.

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

# A Additional Details on Experiments

**Computing Resources**. For most experiments, we use a workstation with a 12-core AMD Ryzen 9 3900X CPU, 64GB RAM, and an NVIDIA Quadro P6000 GPU with 24 GB GPU Memory. BUDDY experiments on ogbl-citation2 require higher CPU RAM, in which case we use a server with 128GB RAM and an NVIDIA A100 GPU with 48 GB GPU Memory.

**Synthetic Graph Generation.** The synthetic graphs are generated by random sampling 10,000 nodes with their features in ogbl-collab [18] and connect 2% of all possible node pairs whose feature similarity falls within specified ranges; all graphs share the same set of nodes and features and only differ in their edges. More specifically, we calculate the pairwise feature similarity between all node pairs and create 50-quantiles of feature similarity scores. We select the 3 smallest quantiles, the 3 largest quantiles, and 4 quantiles in equal intervals in between, resulting in 10 quantiles. We then create 10 synthetic graphs by connecting node pairs whose feature similarity scores fall within the same quantile. Thus, by gradually increasing the range of similarity for connected nodes, we create graphs which resemble different types of link prediction tasks and average feature similarity. We list the average feature similarity score of each synthetic graph in Table 3: the negative extreme is the most negatively correlated graph featuring edges with feature similarity scores ranging in $[-1, -0.33]$, which resembles the heterophilic link prediction task; the positive extreme is the most positively correlated graph with edges similarity scores in $[0.44, 1.00]$, resembling the homophilic link prediction task. Other synthetic graphs correspond to the gated link prediction task with feature similarity scores ranging from the negative to positive spectrum.

Comparing with synthetic graph generation methods in existing literature studying heterophilous graphs, most of these methods focus on node classification tasks. For example, [1] and [60] propose modified preferential attachment processes where edge probabilities are determined by both class compatibility matrices and node degrees, while [7] employs a contextual stochastic block model (CSBM). However, these approaches control homophily/heterophily levels based on node class labels, which are typically unavailable in link prediction settings. More recently, [22] proposed NetInfoF for generating link prediction benchmarks by controlling correlations between node features and edge existence. However, their approach only supports three discrete correlation levels (fully, partially, or uncorrelated) and cannot generate graphs with negatively correlated features among connected nodes. In contrast, our generation process allows for fine-grained control over feature similarity distributions and can produce graphs spanning the full spectrum from negative to positive feature correlations.

**Experiment Setups of the Synthetic Graphs.** For each generated synthetic graph, we randomly split the edges into training, validation, and test sets with a ratio of 8:1:1. We repeat each experiment 3 times with different random seeds and report the average performance with standard deviation in Table 2. We use 2 convolutional layers for SAGE and GCN, and 256 hidden dimensions for all neural network models.

**Additional Experiment Setups of the Real-world Datasets**. We consider three real-world datasets: (1) ogbl-collab [18], a collaboration network between researchers, with nodes representing authors, edges representing co-authorship, and node features as the average word embeddings of the author's papers; (2) ogbl-citation2 [18], a citation network where nodes represent papers and edges represent citations, with the average word embeddings of the paper's title and abstract as node features; (3) e-comm [61], a sparse graph extracted from [37] representing exact matches of queries and related products in Amazon Search, with BERT-embeddings of queries and product information as node features. For ogbl-collab and ogbl-citation2, we follow the recommended metrics (Hit@50 and MRR, respectively) and train-validation-test splits provided by OGB [18]. For e-comm, we use the splits shared by [61], adopt the SpotTarget approach [61] as graph mini-batch sampler for training and use MRR as the evaluation metric. We report each method's average performance on the full test split across 3 runs with different random seeds, and present the results in Table 1.

**Creation of Node Degree and Feature Similarity Buckets on Real-world Datsets**. To create these buckets, we first calculate each edge's feature similarity score and the minimum degree of its two connected nodes. We then create on each graph three buckets per property based on the distribution of feature similarity scores and node degrees, with each bucket covering one-third quantile, except for node degrees of e-comm, where only two buckets are created due to its sparsity.

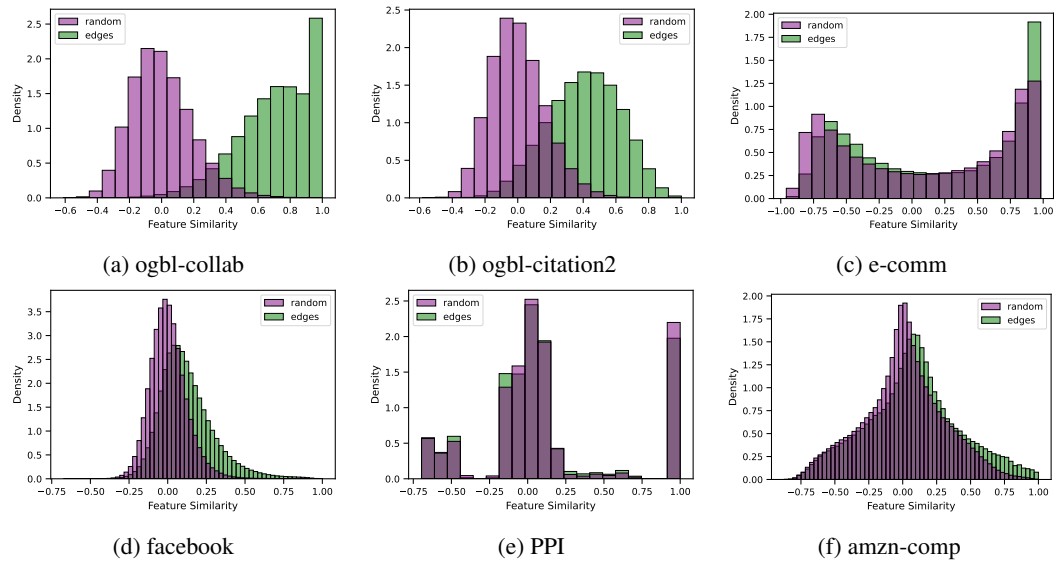

(a) ogbl-collab      (b) ogbl-citation2      (c) e-comm

(d) facebook      (e) PPI      (f) amzn-comp

Figure 6: Comparison of feature similarity distributions for edges and random node pairs on real-world datasets used in our experiments. For similarity scores of random node pairs, we randomly sample 1000 nodes and compute the pairwise cosine similarity between these node features. Similarity score distributions for random node pairs are good approximations of the distributions for non-edge node pairs due to the sparsity of the graphs.

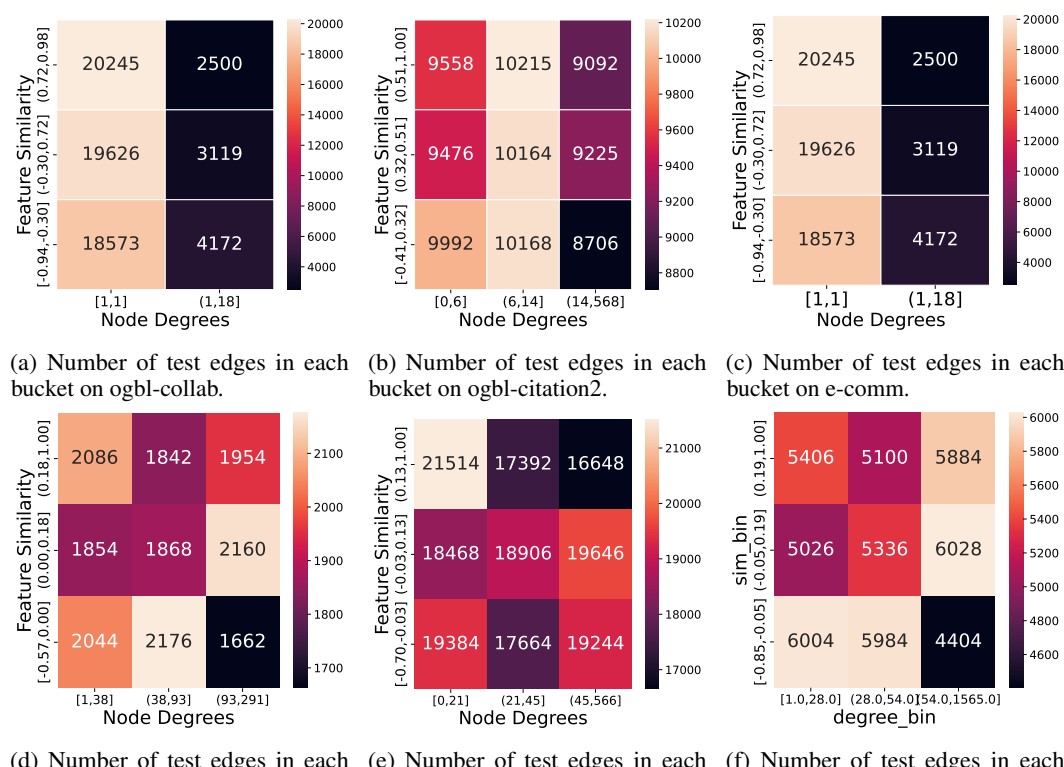

(a) Number of test edges in each bucket on ogbl-collab.

(b) Number of test edges in each bucket on ogbl-citation2.

(c) Number of test edges in each bucket on e-comm.

(d) Number of test edges in each bucket on facebook.

(e) Number of test edges in each bucket on PPI.

(f) Number of test edges in each bucket on amazon-computer.

Figure 7: Number of edges in each bucket on different datasets.

Table 2: Results on synthetic graphs. We report MRR averaged over 3 runs.

| Graph Index | 0 | 1 | 2 | 3 | 4 |
|---|---|---|---|---|---|
| Feat. Sim $K$ | $-0.38_{\pm0.04}$ | $-0.31_{\pm0.01}$ | $-0.28_{\pm0.01}$ | $-0.14_{\pm0.00}$ | $-0.04_{\pm0.00}$ |
| CN | 0.69 | 0.65 | 0.66 | 1.58 | 2.44 |
| RA | 0.69 | 0.65 | 0.66 | 1.64 | 2.58 |
| AA | 0.69 | 0.65 | 0.66 | 1.63 | 2.57 |
| PPR | 10.91 | 3.30 | 2.46 | 3.17 | 3.68 |
| GCN+DOT | $15.88_{\pm0.37}$ | $8.16_{\pm0.11}$ | $5.96_{\pm0.07}$ | $3.55_{\pm0.02}$ | $3.79_{\pm0.00}$ |
| SAGE+DOT | $25.53_{\pm0.09}$ | $9.35_{\pm0.02}$ | $6.32_{\pm0.02}$ | $3.80_{\pm0.07}$ | $3.51_{\pm0.04}$ |
| GCN+DistMult | $46.19_{\pm0.98}$ | $17.20_{\pm0.16}$ | $12.57_{\pm0.05}$ | $4.86_{\pm0.03}$ | $3.58_{\pm0.11}$ |
| SAGE+DistMult | $80.97_{\pm0.32}$ | $23.85_{\pm0.04}$ | $14.66_{\pm0.03}$ | $6.05_{\pm0.08}$ | $6.57_{\pm0.05}$ |
| NoGNN+MLP | $57.84_{\pm0.37}$ | $23.83_{\pm0.08}$ | $18.30_{\pm0.09}$ | $9.05_{\pm0.02}$ | $7.89_{\pm0.04}$ |
| GCN+MLP | $45.69_{\pm0.47}$ | $17.43_{\pm0.08}$ | $12.62_{\pm0.08}$ | $5.03_{\pm0.06}$ | $3.98_{\pm0.06}$ |
| SAGE+MLP | $75.39_{\pm0.64}$ | $43.37_{\pm2.35}$ | $33.64_{\pm1.01}$ | $19.85_{\pm0.68}$ | $23.84_{\pm3.15}$ |
| BUDDY-NoFeat | $27.51_{\pm0.05}$ | $10.12_{\pm0.95}$ | $6.72_{\pm0.85}$ | $2.37_{\pm0.37}$ | $2.15_{\pm0.11}$ |
| BUDDY-GCN | $69.47_{\pm0.65}$ | $24.26_{\pm0.56}$ | $17.60_{\pm0.43}$ | $6.46_{\pm0.10}$ | $4.99_{\pm0.07}$ |
| BUDDY-SIGN | $75.27_{\pm2.33}$ | $26.65_{\pm0.96}$ | $20.81_{\pm0.39}$ | $9.59_{\pm0.29}$ | $7.57_{\pm0.05}$ |
| Graph Index | 5 | 6 | 7 | 8 | 9 |
| Feat Sim $K$ | $0.07_{\pm0.00}$ | $0.24_{\pm0.01}$ | $0.34_{\pm0.01}$ | $0.40_{\pm0.02}$ | $0.54_{\pm0.09}$ |
| CN | 2.81 | 6.26 | 11.97 | 18.62 | 70.49 |
| RA | 2.96 | 6.48 | 12.61 | 20.24 | 77.16 |
| AA | 2.96 | 6.50 | 12.37 | 19.21 | 71.91 |
| PPR | 3.86 | 7.44 | 13.21 | 19.62 | 62.06 |
| GCN+DOT | $4.16_{\pm0.04}$ | $8.11_{\pm0.02}$ | $14.27_{\pm0.05}$ | $20.23_{\pm0.10}$ | $61.57_{\pm0.22}$ |
| SAGE+DOT | $2.29_{\pm0.10}$ | $6.95_{\pm0.08}$ | $14.79_{\pm0.02}$ | $23.95_{\pm0.02}$ | $79.90_{\pm0.13}$ |
| GCN+DistMult | $4.08_{\pm0.01}$ | $7.97_{\pm0.01}$ | $13.99_{\pm0.03}$ | $19.85_{\pm0.08}$ | $62.92_{\pm0.82}$ |
| SAGE+DistMult | $6.71_{\pm0.13}$ | $7.33_{\pm0.08}$ | $16.05_{\pm0.02}$ | $27.52_{\pm0.05}$ | $88.97_{\pm0.35}$ |
| NoGNN+MLP | $8.86_{\pm0.13}$ | $14.78_{\pm0.21}$ | $23.59_{\pm0.32}$ | $31.88_{\pm0.55}$ | $70.55_{\pm0.83}$ |
| GCN+MLP | $4.19_{\pm0.02}$ | $8.12_{\pm0.05}$ | $14.03_{\pm0.02}$ | $20.07_{\pm0.14}$ | $62.24_{\pm0.39}$ |
| SAGE+MLP | $30.35_{\pm0.12}$ | $35.33_{\pm0.75}$ | $49.73_{\pm1.42}$ | $64.38_{\pm0.88}$ | $89.79_{\pm0.74}$ |
| BUDDY-NoFeat | $2.30_{\pm0.08}$ | $5.10_{\pm0.97}$ | $10.54_{\pm2.05}$ | $18.13_{\pm0.04}$ | $67.96_{\pm0.01}$ |
| BUDDY-GCN | $5.39_{\pm0.05}$ | $10.11_{\pm0.45}$ | $19.34_{\pm1.47}$ | $29.44_{\pm2.36}$ | $84.44_{\pm1.22}$ |
| BUDDY-SIGN | $8.45_{\pm0.08}$ | $16.17_{\pm0.44}$ | $27.49_{\pm2.33}$ | $38.39_{\pm4.77}$ | $87.59_{\pm1.40}$ |

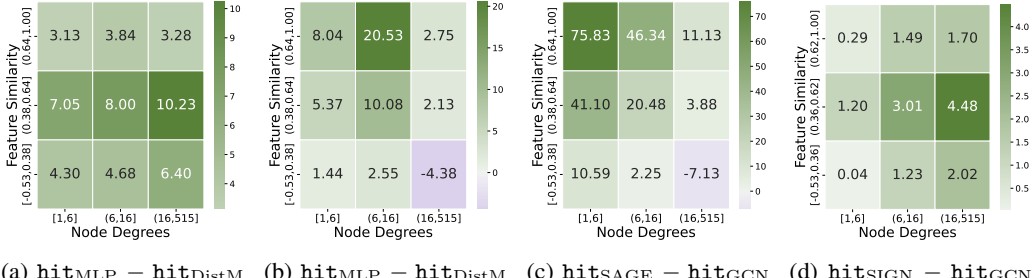

(a) $\text{hit}_{\text{MLP}} - \text{hit}_{\text{DistM}}$ with SAGE encoder.
(b) $\text{hit}_{\text{MLP}} - \text{hit}_{\text{DistM}}$ with GCN encoder.
(c) $\text{hit}_{\text{SAGE}} - \text{hit}_{\text{GCN}}$ with MLP decoder.
(d) $\text{hit}_{\text{SIGN}} - \text{hit}_{\text{GCN}}$ with BUDDY & MLP.

Figure 8: Performance comparison on ogbl-collab among different node degree and edge similarity scores buckets.

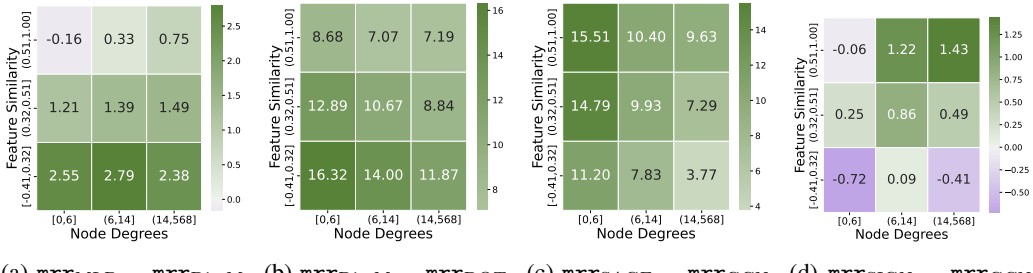

(a) $\mathrm{mrr}_{\mathrm{MLP}} - \mathrm{mrr}_{\mathrm{DistM}}$ with SAGE encoder.

(b) $\mathrm{mrr}_{\mathrm{DistM}} - \mathrm{mrr}_{\mathrm{DOT}}$ with SAGE encoder.

(c) $\mathrm{mrr}_{\mathrm{SAGE}} - \mathrm{mrr}_{\mathrm{GCN}}$ with MLP decoder.

(d) $\mathrm{mrr}_{\mathrm{SIGN}} - \mathrm{mrr}_{\mathrm{GCN}}$ with BUDDY & MLP.

Figure 9: Performance comparison on ogbl-citation2 among different node degree and edge similarity scores buckets.

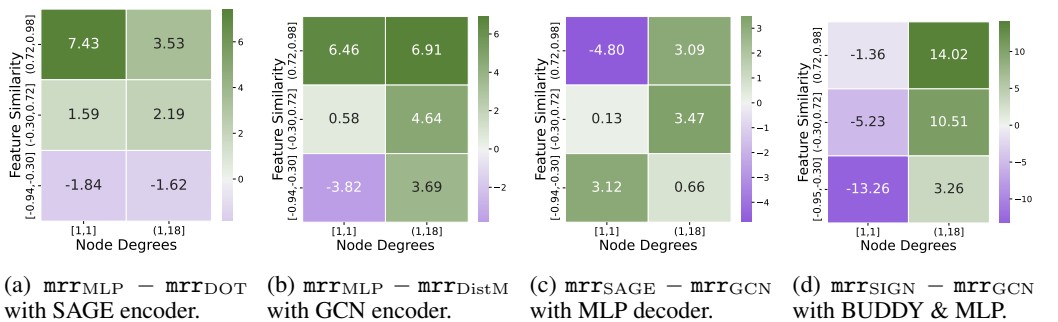

(a) $\mathrm{mrr}_{\mathrm{MLP}} - \mathrm{mrr}_{\mathrm{DOT}}$ with SAGE encoder.

(b) $\mathrm{mrr}_{\mathrm{MLP}} - \mathrm{mrr}_{\mathrm{DistM}}$ with GCN encoder.

(c) $\mathrm{mrr}_{\mathrm{SAGE}} - \mathrm{mrr}_{\mathrm{GCN}}$ with MLP decoder.

(d) $\mathrm{mrr}_{\mathrm{SIGN}} - \mathrm{mrr}_{\mathrm{GCN}}$ with BUDDY & MLP.

Figure 10: Performance comparison on e-comm among different node degree and edge similarity scores buckets.

## B  Proofs of Theorems

### B.1  Proof of Theorem 1

**Proof 1** *We prove separately for homophilic and heterophilic link prediction problems.*

***Homophilic Link Prediction Problem***. *For homophilic link prediction problem, we have $1 \geq \inf(\mathcal{K}_{pos}) = M > \sup(\mathcal{K}_{neg})$. As $M = \inf(\mathcal{K}_{pos})$, a pair of nodes $u, v \in \mathcal{V}$ must exist in the training graph with feature vectors $\mathbf{x}_u = (\cos\theta_u, \sin\theta_u)$ and $\mathbf{x}_v = (\cos\theta_v, \sin\theta_v)$ such that $k(u, v) = \cos(\theta_u - \theta_v) = M$.*

*Now let us consider a fully optimized DistMult decoder for the problem. We parameterize the DistMult decoder with $\mathbf{w} = (w_1, w_2)$ and $b$; in this case, the predicted link probability*

$$\hat{y}_{uv} = (\mathbf{x}_u \otimes \mathbf{x}_v)^\mathsf{T} \mathbf{w} + b = w_1 \cos\theta_u \cos\theta_v + w_2 \sin\theta_u \sin\theta_v + b$$

*As $M = \inf(\mathcal{K}_{pos})$, any pairs of nodes $(u', v')$ that has feature similarity $k(u', v')$ slightly smaller than $M$ should have $\hat{y}_{u'v'} \leq 0$. Therefore, we must have $\hat{y}_{uv} = 0$. Furthermore, since we are bounding $\hat{y}_{u'v'} = 1$ if $k(u', v') = 1$ during training, we must have $\hat{y}_{uu} = 1$ and $\hat{y}_{vv} = 1$. Using $\hat{y}_{uv} = 0$, $\hat{y}_{uu} = 1$ and $\hat{y}_{vv} = 1$, we can obtain the solutions for $w_1$, $w_2$ and $b$ as follows:*

$$w_1 = \frac{1}{1 - \cos(\theta_u - \theta_v)}, w_2 = \frac{1}{1 - \cos(\theta_u - \theta_v)}, b = \frac{1}{\cos(\theta_u - \theta_v) - 1} + 1$$

*As $w_1 = w_2 = \frac{1}{1-M}$, we have the similarity score for arbitrary node pair $(u', v')$ as*

$$\hat{y}_{u'v'} = \frac{1}{1 - M} \cos(\theta_{u'} - \theta_{v'}) + \frac{1}{M - 1} + 1 = \frac{1}{1 - M} k(u', v') + \frac{1}{M - 1} + 1$$

*We can verify that the above link prediction model is fully optimized with loss function $\mathcal{L} = 0$ as it always yield $\hat{y}_{uv} \geq 0$ for $k(u', v') \geq M$ and $\hat{y}_{uv} < 0$ for $k(u', v') < M$. Therefore, we show that $\hat{y}_{u'v'}$ increases with $k(u', v')$ at a linear rate of $\frac{1}{(1-M)}$ for homophilic link prediction problem.*

***Heterophilic Link Prediction Problem***. *For heterophilic link prediction problem, we have $-1 \leq \sup(\mathcal{K}_{pos}) = M < \inf(\mathcal{K}_{neg})$. As $M = \sup(\mathcal{K}_{pos})$, a pair of nodes $u, v \in \mathcal{V}$ must exist in the training graph with feature vectors $\mathbf{x}_u = (\cos\theta_u, \sin\theta_u)$ and $\mathbf{x}_v = (\cos\theta_v, \sin\theta_v)$ such that $k(u, v) = \cos(\theta_u - \theta_v) = M$.*

*Similar to the homophilic case, as $M = \sup(\mathcal{K}_{pos})$, any pairs of nodes $(u', v')$ that has feature similarity $k(u', v')$ slightly larger than $M$ should have $\hat{y}_{u'v'} \leq 0$. Therefore, we must have $\hat{y}_{uv} = 0$. Furthermore, since we are bounding $\hat{y}_{u'v'} = 1$ if $k(u', v') = -1$ during training, for nodes $u*$ and $v*$ where $\mathbf{x}_{u*} = -\mathbf{x}_u$ and $\mathbf{x}_{v*} = -\mathbf{x}_v$, we must have $\hat{y}_{uu*} = 1$ and $\hat{y}_{vv*} = 1$. Using $\hat{y}_{uv} = 0$, $\hat{y}_{uu*} = 1$ and $\hat{y}_{vv*} = 1$, we can obtain the solutions for $w_1$, $w_2$ and $b$ as follows:*

$$w_1 = -\frac{1}{1 + \cos(\theta_u - \theta_v)}, w_2 = -\frac{1}{1 + \cos(\theta_u - \theta_v)}, b = 1 - \frac{1}{1 + \cos(\theta_u - \theta_v)}$$

*As $w_1 = w_2 = -\frac{1}{1+M}$, we have the similarity score for arbitrary node pair $(u', v')$ as*

$$\hat{y}_{u'v'} = -\frac{1}{1 + M}\cos(\theta_{u'} - \theta_{v'}) + 1 - \frac{1}{1 + M} = -\frac{1}{1 + M}k(u', v') + 1 - \frac{1}{1 + M}$$

*We can similarly verify that the above link prediction model is fully optimized with loss function $\mathcal{L} = 0$ as it always yield $\hat{y}_{uv} \geq 0$ for $k(u', v') \leq M$ and $\hat{y}_{uv} < 0$ for $k(u', v') > M$. Therefore, we show that $\hat{y}_{u'v'}$ decreases with $k(u', v')$ at a linear rate of $\frac{1}{(1-M)}$ for heterophilic link prediction problem.* ∎

## B.2 Proof of Theorem 2

**Proof 2** *The decision boundary for gated link prediction is non-linear since it involves multiple disjoint intervals for each class (edge and non-edge). A linear model can only create a single threshold $x_0$ to separate the classes: one region for $x \leq x_0$ and the other for $x > x_0$. Therefore, there always exists misclassified points regardless of the choice of $x_0$.* ∎

## B.3 Proof of Theorem 3

**Proof 3** *We begin by considering the decoder trained with the linear GNN representation $\mathbf{r}_u$ that does not separate ego- and neighbor-embeddings in its message passing. Following our assumptions, the aggregated representation $\mathbf{r}_1$ for training nodes with feature $\mathbf{x}_1$ is $\mathbf{r}_1 = \frac{1}{d+1}\mathbf{x}_1 + \frac{d}{d+1}\mathbf{x}_2$, and $\mathbf{r}_2$ can be obtained similarly.*

*Similar to Proof 1, we parameterize the DistMult decoder with $\mathbf{w} = (w_1, w_2)$ and $b$. Suppose node $u$ has feature vector $\mathbf{x}_1$ and node $v$ has feature vector $\mathbf{x}_2$. As the training graph is heterophilic and $(u, v)$ is connected on the graph, we should have $\hat{y}_{uv} > 0$ in this case. Without the loss of generality, we set $\hat{y}_{uv} = 1$. Furthermore, as self-loops $(u, u)$ and $(v, v)$ are not connected in the graph, we should also have $\hat{y}_{uu} = \alpha < 0$ and $\hat{y}_{vv} = \alpha < 0$. Using $\hat{y}_{uv} = 1$, $\hat{y}_{uu} = \alpha < 0$ and $\hat{y}_{vv} = \alpha < 0$, we can obtain the solutions for $w_1$, $w_2$ and $b$ as follows:*

$$w_1 = w_2 = -\frac{(\alpha - 1)(d + 1)^2}{(d - 1)^2(\cos(\theta_1 - \theta_2) - 1)}$$

$$b = \frac{2\sin(\theta_1 + \theta_2)\left(-d^2 + (\alpha + d(\alpha d - 2))\cos(\theta_1 - \theta_2) + 2\alpha d - 1\right)}{(d - 1)^2(-2\sin(\theta_1 + \theta_2) + \sin(2\theta_1) + \sin(2\theta_2))}$$

*Now we apply the optimized DistMult decoder on the test nodes with degree $d'$. For the test node $u'$ with feature vector $\mathbf{x}_1$, the aggregated representation $\mathbf{r}'_1$ is $\mathbf{r}'_1 = \frac{1}{d'+1}\mathbf{x}_1 + \frac{d'}{d'+1}\mathbf{x}_2$. Similarly, $\mathbf{r}'_2$ can be obtained.*

*Assume the test nodes $u'$ has feature vector $\mathbf{x}_1$ and node $v'$ has feature vector $\mathbf{x}_2$. Based on assumptions, $(u', v')$ should be connected in the heterophilic graph. Plugging in the optimized DistMult parameters, the predicted link probability $\hat{y}_{u'v'}$ for positive (edge) node pairs can be written as*

$$\hat{y}_{u'v'} = \frac{2\alpha\left(d - d'\right)\left(dd' - 1\right) - 4dd' + \left(d'\right)^2 + d^2\left(\left(d'\right)^2 + 1\right) + 1}{(d - 1)^2\left(d' + 1\right)^2}$$

*On the other hand, for nodes with the same feature vectors (e.g., self-loops), they should not be connected in the graph. The predicted link probability $\hat{y}_{u'u'}$ for self-loops can be written as*

$$\hat{y}_{u'u'} = \frac{\alpha \left( -4dd' + (d')^2 + d^2 \left( (d')^2 + 1 \right) + 1 \right) + 2 \left( d - d' \right) \left( dd' - 1 \right)}{(d-1)^2 \left( d' + 1 \right)^2}$$

*Thus, the separation distance between edges and non-edges for the test nodes on DistMult decoder optimized with GNN representations can be represented as*

$$\Delta_{\mathrm{GNN}} = |\hat{y}_{u'v'} - \hat{y}_{u'u'}|.$$

*Now let us consider the baseline DistMult decoder optimized with node features $\mathbf{x}_u$. It is straight forward to see that if the decoder is optimized such that $\hat{y}_{uv} = 1$, $\hat{y}_{uu} = \alpha < 0$ and $\hat{y}_{vv} = \alpha < 0$, we will continue to have $\hat{y}_{u'v'} = 1$ and $\hat{y}_{u'u'} = \alpha$ for the test nodes, as the decoder is graph-agnostic. Thus, the separation distance between edges and non-edges for the test nodes on the baseline DistMult decoder can be represented as $\Delta_{\mathrm{baseline}} = 1 - \alpha$.*

*The GNN-based DistMult decoder reduces the separation distance between edges and non-edges for the test nodes compared to the baseline DistMult decoder when $\Delta_{\mathrm{GNN}} < \Delta_{\mathrm{baseline}}$. Solving this inequality for integers $d$ and $d'$ under the constraints of $d \geq 0, d' \geq 0, \alpha < 0$, we obtain the solutions as $d' > 0$ if $d = 0$, or $1 \leq d' < d$ if $d \geq 2$.* ∎

## C   Additional Real-World Datasets Exhibiting Feature Heterophily

In addition to the dataset we employ in experiments, we identified a diverse range of heterophilious datasets from other graph learning tasks (e.g. graph or node classification). Specifically, we have identified a range of biological graph datasets from the TUDataset [31] that exhibit feature heterophily (these datasets are typically used for graph classification). In particular, the following datasets comprise entirely heterophilic graphs (i.e., every single graph has negative homophily ratios):

- aspirin
- benzene
- malonaldehyde
- naphthalene
- salicylic_acid
- toluene
- uracil

These datasets are substantial in size, as detailed below:

Table 3: Number of entirely heterophilic Graphs in Different Datasets (from TUDataset)

| Dataset | Number of Graphs |
|---|---|
| aspirin | 111,763 |
| benzene | 527,984 |
| malonaldehyde | 893,238 |
| naphthalene | 226,256 |
| salicylic_acid | 220,232 |
| toluene | 342,791 |
| uracil | 133,770 |

In addition, some datasets contain instances that are strongly heterophilious (with homophily ratio -1.0), including bbbp, NCI1, AIDS, and QM9 [45]. Note that such findings are not only limited to biological datasets. For instance, 73% of graphs from PATTERN [13] (Mathematical Modeling) have negative homophily ratios.

Furthermore, we have identified node classification benchmarks from torch-geometric.datasets that display a wide range of feature homophily ratios, many of which are more heterphilious than e-comm, the one we proposed in paper. Notably, some real-world benchmarks exhibit negative homophily ratios. We summarize our findings in Table 4.

In addition to the homophily ratios presented above, we provide the feature similarity distributions for edges and random node pairs across several datasets in Figure 11, following the same convention as Figure 5 in our paper. These plots reveal clear signs of heterophily in the existing benchmark graphs.

Table 4: Homophily Ratios for Different Datasets

| Dataset | Homophily Ratios |
| --- | --- |
| Ogbl-ppa | 0.74 |
| Ogbl-collab | 0.70 |
| Ogbl-citat2 | 0.40 |
| WikiCS | 0.35 |
| PubMed [4] | 0.22 |
| e-comm | 0.18 |
| DBLP [4] | 0.13 |
| Cora [4] / FacebookPagepage [39] | 0.12 |
| AQSOL [13] / Yelp | 0.12 |
| PPI [63] | 0.11 |
| Facebook [48] | 0.11 |
| Amazon-Photo [40] | 0.10 |
| Amazon-Computers [40] | 0.07 |
| Twitch-DE [39] | 0.07 |
| Twitch-FR [39] | 0.06 |
| BlogCatalog [48] | 0.06 |
| CiteSeer [48] | 0.05 |
| TWeibo [48] | 0.01 |
| Karateclub [50] | -0.03 |
| UPFD [11] | -0.10 |
| BBBP instances [45] | -1.00 |
| NC11 instances [31] | -1.00 |
| AIDS instances [31] | -1.00 |
| QM9 instances [45] | -1.00 |

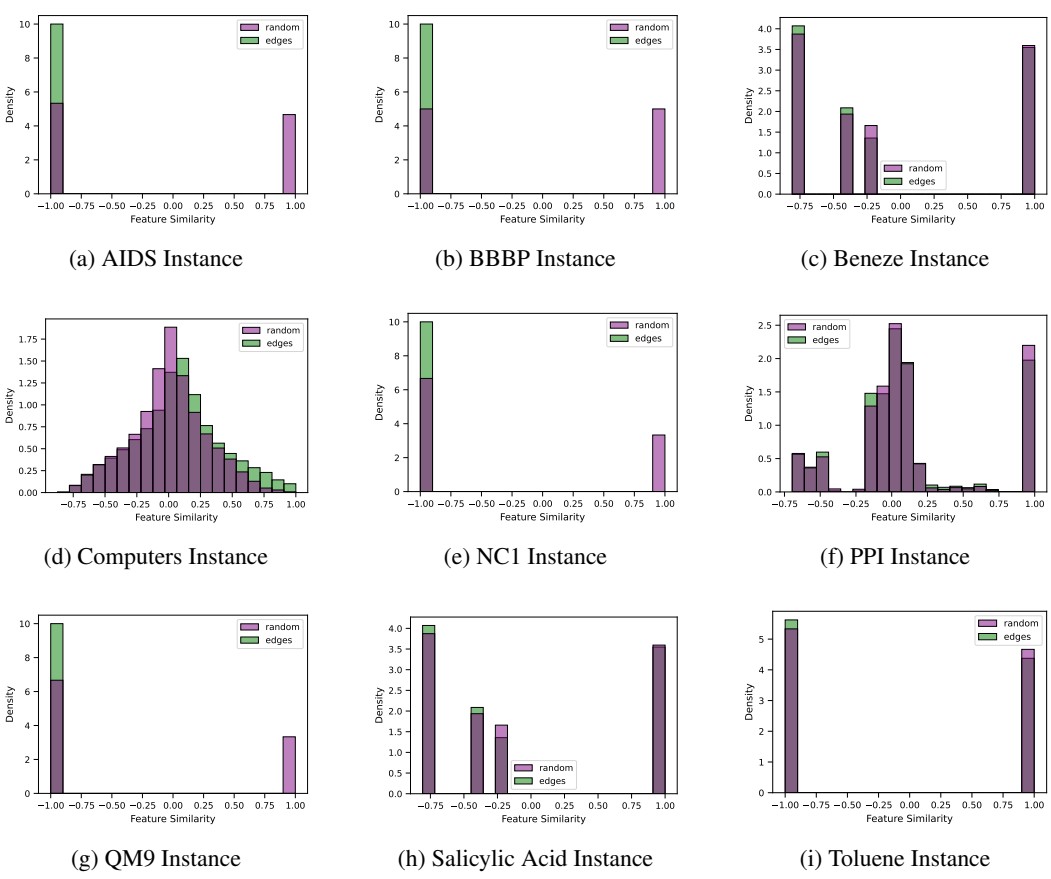

Figure 11: Heterophilic Sample Feature Distributions

