# OpenReview forum: "On the Impact of Feature Heterophily on Link Prediction with Graph Neural Networks"
_NeurIPS.cc/2024/Conference — NeurIPS 2024 poster_

### Official Review · Reviewer_ufZT · 2024-06-25

**Soundness:** 3
**Presentation:** 4
**Contribution:** 2
**Rating:** 4
**Confidence:** 5

**Summary:**

This paper studies the impact of feature heterophily, rather than class heterophily, on link prediction tasks with GNN. It introduces the definition of non-homophilic link prediction based on the pairwise feature similarity of connected/non-connected node pairs of the graph. It further shows that the choice of encoders (GCN vs SAGE) and decoders (linear vs MLP) can lead to different link prediction performance on non-homophilic graphs. Various experiments on both synthetic and real graph benchmarks showcase that the feature heterophily can influence the GNNs' performance on link prediction tasks.

**Strengths:**

1. The paper writing is clear and easy to understand.

2. The research problem of feature heterophily on link prediction tasks is novel.

3. The theoretical analysis part is comprehensive and intuitive.

4. The experiment is interesting, especially the comparisons of encoders/decoders on graphs with different levels of feature heterophily. It supports the claim made in the analysis part.

**Weaknesses:**

1. While the research problem of feature heterophily is novel, the significance of the study is limited. The authors propose the concept of Heterophilic Link Prediction tasks. However, it is almost impossible to find such graphs in the real world. The most non-homophilic graph in the experiment is E-Commerce, which also has positive feature similarity. It will make the study more significant if the authors provide more real-world graphs with strong Heterophilic properties. Otherwise, it makes the study less practical and the Heterophilic Link Prediction becomes an *artificial* problem.

2. To consolidate the research problem and show that the synthetic graphs are valid in the real world, there can be some discussions about how the Synthetic Graph Generation is related to the other random graph generation methods.

3. While the research is centered around GNNs, Sec 3 and Sec 4.1 have no clear connection to GNNs. Also, Theorem 1 and 2 have no connection to GNNs. The discussion brings nothing beyond classical ML that a classification problem with complex decision boundaries requires a model with higher expressiveness to fit in.

**Questions:**

1. Beyond the graphs with node features, is there any insight from this study on graphs without node features? Non-attributed graphs are a significant set of benchmarks for link prediction tasks.

2. Beyond the feature heterophily in the direct neighbors, are there any discussions or analyses about the feature heterophily of k-hop neighbors? I suspect in the real-world graphs, k-hop heterophily is much more common, which can further improve the significance of this study.

3. Is it possible to expand the scope of this study to any link prediction method rather than GNNs?

**Limitations:**

See the weakness.

---

> ### Author Rebuttal · Authors · 2024-08-07
>
> **W1 Significance of the work**
>
> Thank you for your feedback. We want to point out that going beyond the global characterization of the graphs for the link prediction task, we also provided a zoom-in analysis on the performance on heterophilous edges per method for all the real-world datasets, which also demonstrates the effectiveness of our identified designs in improving GNN performance on the full spectrum of low-to-high feature similarity (c.f. Line 360-365, 370-374, and Figure 4). This is important since in reality every dataset exhibits variance in feature similarity across all the links. That said, we acknowledge that finding real-world, *publicly-available* graphs that align with the heterophilic link prediction task, as discussed in Section 3, is non-trivial. Currently, we have identified e-comm as a high-quality non-homophilic network for link prediction. In Figure 1 of our general response plots, we have demonstrated that many real-world heterophilous node classification benchmarks, such as Pokec [1] and Amazon Ratings [2], are predominantly homophilous for the link prediction task, with most edges connecting nodes with similar features. Nevertheless, as we mentioned above, these datasets still have heterophilous edges where our theory and findings apply.
>
> We believe that the current limited number of high-quality heterophilous link prediction datasets does not imply that the problem itself is artificial. To bring in a bit more context, the lack of high-quality heterophilous datasets for *node classification* tasks was also a longstanding challenge before the challenge of heterophily for GNNs became widely recognized in the field. As the problem attracted more interest, various research teams dedicated their efforts to introducing a range of heterophilous datasets, thereby advancing findings and methodologies within the subfield. Additionally, as we mentioned above, heterophilous connections can also arise *locally* even if the graph is homophilous in general, which motivates our locallized, more nuanced analysis (cf. Figures 4 and 9).  Thus, by casting light to the problem of link prediction beyond homophily, we hope that our work will motivate the introduction of a high-quality set of benchmarks with diverse feature similarity, beyond the real data that we presented.
>
> [1] Large Scale Learning on Non-Homophilous Graphs: New Benchmarks and Strong Simple Methods. (NeurIPS 2021).
>
> [2] A critical look at the evaluation of GNNs under heterophily: Are we really making progress? (ICLR 2022).
>
> **W2 Relation of synthetic graph generation with other random graph generation methods**
>
> Our synthetic graph generation process is detailed in Lines 512-520 in Appendix A. Most of the random graph generation methods in existing literature studying heterophilous graphs are for creating synthetic datasets for node classification: [3,4] follow a modified preferential attachment process, where the probability of an edge is determined by both the class compatibility matrix and the node degree, while [5] follows to use contextual stochastic block model (CSBM) for synthetic graph generation. However, all of the aforementioned approaches control the homophily/heterophily levels defined on class labels of the generated synthetic graphs, while for graphs for link prediction usually do not have class labels. NetInfoF [6] recently generates synthetic graphs as link prediction benchmarks by controlling the correlation between node feature and edge existence. However, their generation process only allows node features to be fully correlated, partially correlated, or uncorrelated with edge existence; it lacks the capability to granularly control the feature similarity of the generated graph as in our graph generation process, and it is also not capable of generating graphs where features are *negatively* correlated among connected nodes. We will include these discussions in our final version.
>
> [3] Mixhop: Higher-order graph convolutional architectures via sparsified neighborhood mixing. (ICML 2019).
>
> [4] Beyond Homophily in Graph Neural Networks: Current Limitations and Effective Designs. (NeurIPS 2020).
>
> [5] Adaptive universal generalized pagerank graph neural network. (arXiv 2020).
>
> [6] NetInfoF Framework: Measuring and Exploiting Network Usable Information. (ICLR 2024).
>
> **W3 Theorem connections with GNNs**
>
> Thank you for the feedback. Existing GNN4LP methods are composed of encoders and decoders, both of which are essential components. Our first two theorems primarily consider the effectiveness of link prediction decoders, which, as our experiment shows, have a significant impact on GNN performance under link prediction tasks. Furthermore, the choice of decoders in link prediction tasks is not as trivial as it might seem: while MLP decoder is more preferred in academic research, DOT product decoder is more widely adopted in industry due to its scalability in tasks such as retrieval. Our theoretical analysis reveals the limitation of DOT product decoder for non-homophilic link prediction tasks and helps identify DistMult as an effective substitute which maintains the scalability of DOT product with closer-to-MLP link prediction accuracy. We will make the connections more clear in our paper.
>
> **Q1 Insights for non-attributed graphs**
>
> Our study primarily focuses on the impact of feature heterophily. As we define heterophily on node features, our definition and analysis is not applicable to graphs without node features. Given the prevalence of attributed graphs in real-world applications (e.g., textual information is often associated with nodes in industrial and other applications), we believe that the contributions of this work are significant for these commonplace settings.
>
> **Q2 Expand the analyses to k-hop neighbors**
>
> We appreciate the reviewer for sharing this valuable suggestion. We believe it would be an interesting direction to explore in future work.

---

> ### Author Response · Authors · 2024-08-07
>
> **Q3 Expand beyond GNN**
>
> Currently, GNN4LP methods are the state-of-the-art methods for the link prediction task; additionally, many heuristics such as common neighbors do not make use of node features. Therefore, we primarily restrict our study to the scope of GNNs. However, the notion of "feature heterophily" can have implications beyond GNNs, such as decoder-only models that only consider pairwise node features. Furthermore, there could be interesting dynamics between feature and structural proximity, which is beyond the scope of this paper and could be an interesting direction for future work. We will mention some of these connections in the paper.

---

> > ### Comment · Reviewer_ufZT · 2024-08-11
> >
> > Thanks for addressing my questions in the rebuttal. However, I still have my main concerns as in the original review:
> >
> > 1. **One real-world dataset is not enough**: I appreciate the author's comments about the high-quality non-homophilic network for link prediction. I agree that finding such datasets was also a challenge when researchers started noticing heterophily issues in node classification tasks. In those studies, there are at least more than one heterophilic real-world datasets being evaluated. It can indicate that such heterophilic issues exist widely in real-world datasets. However, this study only finds one such dataset with heterophilic properties. This greatly undermines the generalization of this study. Researchers will be hardly interested in novel research that is limited to only one real-world dataset. On the other hand, if the paper can present more heterophilic datasets, it can enhance the contribution of this study, and have a huge impact on the research community just like heterophily issues in node classification. In this way, this problem can attract more interest.
> >
> > 2. **Method part is trivial**: If we look into the method part of the paper, the proposed method is trivial. If we look at the heterophilic papers the authors suggested, they presented novel methods even though evaluating heavily on the synthetic datasets. So another way to improve the study is to revise the method part, which can introduce more contribution to the research value.
> >
> > Due to the main concerns above, I keep my original rating.

---

> ### Author Response · Authors · 2024-08-12
> **Response to Reviewer ufZT (1)**
>
> Thank you for your additional feedback. We have carefully considered your concerns and would like to address them as follows:
>
> 1. Following your suggestion, we have conducted a thorough examination of various real-world benchmarks, including those that are typically used for other graph learning tasks (e.g., graph or node classification). We see the challenge of identifying heterophilic link prediction benchmarks as an opportunity to introduce more diverse, real datasets for this task. Specifically, we have identified a range of biological graph datasets from the TUDataset [1] that exhibit feature heterophily (these datasets are typically used for graph classification). In particular, the following datasets comprise **entirely heterophilic graphs** (i.e., every single graph has negative homophily ratios):
>
> - aspirin
> - benzene
> - malonaldehyde
> - naphthalene
> - salicylic_acid
> - toluene
> - uracil
>
> These datasets are substantial in size, as detailed below:
>
> | Dataset (from TUDataset) | Number of Graphs |
> |--------------------------|------------------|
> | aspirin                   | 111,763           |
> | benzene                   | 527,984           |
> | malonaldehyde             | 893,238           |
> | naphthalene               | 226,256           |
> | salicylic_acid            | 220,232           |
> | toluene                   | 342,791           |
> | uracil                    | 133,770           |
>
> In addition, some datasets contain instances that are **strongly heterophilious** (with homophily ratio -1.0), including bbbp, NCI1, AIDS, and QM9 [2]. Note that such findings are not only limited to biological datasets. For instance, 73% of graphs from PATTERN [3] (Mathematical Modeling) have negative homophily ratios.
>
> Furthermore, we have identified node classification benchmarks from torch-geometric.datasets that display a wide range of feature homophily ratios, many of which are more heterphilious than e-comm, the one we proposed in paper. Notably, some real-world benchmarks exhibit negative homophily ratios. We summarize our findings in the following table.
>
> | Dataset                                   | Homophily Ratios |
> | ----------------------------------------- | ---------------- |
> | Ogbl-ppa                                  | 0.74             |
> | Ogbl-collab (in the paper)                             | 0.70             |
> | Ogbl-citat2  (in the paper)                             | 0.40             |
> | WikiCS [4]                                | 0.35             |
> | PubMed [5]                                | 0.22             |
> | e-comm (in the paper)                                   | 0.18             |
> | DBLP   [5]                                | 0.13             |
> | Cora   [5] / FacebookPagepage [10]        | 0.12             |
> | AQSOL  [6] / Yelp                         | 0.12             |
> | PPI    [7]                                | 0.11             |
> | Facebook [8]                              | 0.11             |
> | Amazon-Photo [9]                          | 0.10             |
> | Amazon-Computers [9]                      | 0.07             |
> | Twitch-DE [10]                            | 0.07             |
> | Twitch-FR [10]                            | 0.06             |
> | BlogCatalog [8]                           | 0.06             |
> | CiteSeer  [8]                             | 0.05             |
> | TWeibo [8]                                | 0.01             |
> | Karateclub [11]                           | -0.03            |
> | UPFD  [12]                                | -0.10            |
> | BBBP instances  [2]                                | -1.00            |
> | NCI1 instances  [1]                                | -1.00            |
> | AIDS instances  [1]                               | -1.00            |
> | QM9 instances  [2]                                | -1.00            |
>
> In addition to the homophily ratios presented above, we provide the feature similarity distributions for edges and random node pairs across several datasets in this [anoynomized GitHub repo](https://anonymous.4open.science/r/FeatureHeteroPlots-D591), following the same convention as Figure 5 in our paper. These plots reveal clear signs of heterophily in the existing benchmark graphs.
>
> Consequently, we believe that our study is grounded in existing benchmarks. We will include these findings in our paper to present a set of diverse link prediction datasets and we will also encourage the introduction of additional heterophilic datasets, thereby broadening the scope of research in this area. Thank you again for leading us to this direction!

---

> ### Author Response · Authors · 2024-08-12
> **Response to Reviewer ufZT (2)**
>
> 2.  Thank you for your suggestion. As a pioneering study in this area, our primary objective has been to characterize the problem and identify useful design principles. We believe that our work is significant in setting a new research direction, which is essential for advancing the field. Furthermore, it is also noteworthy that many complex architecture designs proposed for handling heterophily in _node classification_ were later found to be not as effective as following the simple principle of separating ego-and neighbor-embeddings in model architecture (e.g., applying it to GAT and Graph Transformer), as highlighted in [13]. Our study takes a comprehensive approach, laying the groundwork for future research. We acknowledge that there is ample room for exploring more effective methods, and we view this as a key avenue for future research.
>
> [1] TUDataset: A collection of benchmark datasets for learning with graphs
>
> [2] MoleculeNet: A Benchmark for Molecular Machine Learning
>
> [3] Benchmarking Graph Neural Networks
>
> [4] Wiki-CS: A Wikipedia-Based Benchmark for Graph Neural Networks
>
> [5] Deep Gaussian Embedding of Graphs: Unsupervised Inductive Learning via Ranking
>
> [6] Benchmarking Graph Neural Networks
>
> [7] Predicting multicellular function through multi-layer tissue networks
>
> [8] PANE: scalable and effective attributed network embedding
>
> [9] Pitfalls of Graph Neural Network Evaluation
>
> [10] Multi-scale Attributed Node Embedding
>
> [11] An Information Flow Model for Conflict and Fission in Small Groups
>
> [12] User Preference-aware Fake News Detection
>
> [13] A critical look at the evaluation of GNNs under heterophily: Are we really making progress?

---

> ### Author Response · Authors · 2024-08-13
> **Response to Reviewer ufZT (3): results on additional heterophilic dataset**
>
> We have additionally conducted experiments on one heterophilic dataset we identified (Amazon-Computers, feature homophily ratio=0.07) and present the result in the table below. We find that the conclusions from our main paper still hold (SAGE > GCN, due to the sparation of the ego- and neighbor-embeddings, and the use of DistMult/MLP decoders is better than DOT).
>
> | Encoder | Decoder  | MRR   | std  |
> | ------- | -------- | ----- | ---- |
> | GCN     | DOT      | 24.67 | 0.61 |
> | GCN     | DistMult | 31.59 | 1.04 |
> | GCN     | MLP      | 52.90 | 0.86 |
> | SAGE    | DOT      | 42.26 | 0.54 |
> | SAGE    | DistMult | 58.42 | 0.12 |
> | SAGE    | MLP      | 58.38 | 0.07 |
> | (NoGNN) | MLP      | 21.15 | 0.23 |
>
> We will include these new results in the final version of the paper.

---

### Official Review · Reviewer_YTzD · 2024-07-10

**Soundness:** 3
**Presentation:** 3
**Contribution:** 3
**Rating:** 7
**Confidence:** 4

**Summary:**

This paper proposes to connect the feature similarity/dissimilarity to the linking possibilities  between node pairs. It theoretically demonstrates the necessity of considering the overall feature heterophily  of a graph for better link prediction by using a two-dimensional model. On the other hand, the numerical experiments are conducted to show the impact of feature heterophily on link prediction and the influences of encoder and decoder in final performance.

**Strengths:**

1. The isssue of  feature heterophily is valuable to be addressed, which can contribute to  the understanding the  formation of links among nodes. This work provides a concise theoretical analysis regarding it.
2. The paper is well-written and easy to follow.
3. The numerical experiments have been done to validate the theoretical results. The results are comprehensive and seems convincing to me.

**Weaknesses:**

1. Since GNN encode node contents and local structures, it is worthwhile to see the relationship between feature similarity and structural similarity, i.e. whether these two properties are consistent, and its role in affecting the performances of heuristic methods and feature learning methods, e.g., the performance comparison on ogbl-collab for two types of link prediction approaches in Table 1.
2. More benchmark datasets can be considered to cover a diversity of feature similarity distribution.

**Questions:**

1. In eq.(1), \hat{y} is the predicted link probability, so \hat{y}>=0. But in Sec.3.2 in theorem assumption, \hat{y} can be negative. I'm confused.

2. The green line in Fig.1 denotes the score distribution of positive node pairs, why all the scores for positive pairs are identical?

3. Theorem 1 is derived based on the assumption that nodes are 2-dimensionl. For high dimensional features, the learning rate is still linearly with 1/(1-M)? The generality of theoretical results is required to discuss.

4. What is the split for training/validation/test on real-world datasets? How does the size of  training data affect the performance of heuristic methods and learning based methods, respectively?
5. I cannot understand why the buckets are divided according to the minimum degree of connected nodes (in x-axis dimension). Why not consider the degree differences or some functions of degrees of two nodes? At least on the surface, nodes with similar degrees have big chance to be linked in assortative graphs, and vice versa in dissassortiative graphs.

**Limitations:**

As pointed out in Questions 3, the theoretical results for more general cases need to be addressed.

---

> ### Author Rebuttal · Authors · 2024-08-07
>
> **W1 Exploring the relationship between feature similarity and structural similarity**
>
> Thank you for raising this valuable suggestion. Exploring the dynamics between feature and structural similarity would indeed be an interesting direction for future research. In this work, our aim is to provide the first essential characterizations of the impact of feature heterophily on link prediction for GNNs. Future work can explore more fine-grained questions, such as the insightful one you raised.
>
> **W2 Benchmark datasets**
>
> Thank you for your feedback. First, we would like to emphasize that we have run experiments on a series of synthetic graphs of different heterophily ratios/feature similarity distributions. Second, going beyond the global characterization of the graphs for the link prediction task, we also provided a zoom-in analysis on the performance on heterophilous edges per method for each real-world dataset, which also demonstrates the effectiveness of our identified designs in improving GNN performance on the full spectrum of low-to-high feature similarity (c.f. Line 360-365, 370-374, and Figure 4). This is important since in reality every dataset exhibits variance in feature similarity across all the links. Finally, by casting light on the problem of link prediction beyond homophily, we hope that our work will motivate the introduction of a high-quality set of benchmarks with diverse feature similarity, beyond the real data that we presented. This trend was also observed in the literature after the challenges of heterophily for GNNs were pointed out in the node classification task [1,2]: as the problem attracted more interest, various research teams dedicated their efforts to introducing a range of heterophilous datasets, thereby advancing findings and methodologies within the subfield. In Figure 1 of our general response plots, we have shown that many real-world heterophilous *node classification* benchmarks, such as pokec [1] and Amazon ratings [2], are homophilous for link prediction, with most edges connecting nodes with similar features. Nevertheless, as we mentioned above, these datasets still have heterophilous edges where our theory and findings apply.
>
> [1] Large Scale Learning on Non-Homophilous Graphs: New Benchmarks and Strong Simple Methods. (NeurIPS 2021).
>
> [2] A critical look at the evaluation of GNNs under heterophily: Are we really making progress? (ICLR 2022).
>
> **Q1 Clarification on \hat{y}**
>
> Thank you for carefully reading our paper! For our theoretical analysis in Sections 3 & 4, the "predicted link probability" $\hat{y}_{u,v}$ is derived based on the equation in Line 144 ("Theoretical Assumptions" paragraph) in Section 3.2.
>
> We note that we should have used "predicted link score" instead of "predicted link probability" for the discussions in Section 3, as the "link probability" term would imply that our prediction $\hat{y}_{u,v}$ is bounded within $[0,1]$ which is not the case in our analysis. We will revise the usage of "predicted link probability" term and make changes to ensure the notations are consistent and clear in our final version.
>
> **Q2 Identical scores for positive pairs**
>
> Thank you for the question. Figure 1 gives some *example distributions* of positive node pairs (i.e., edges – colored in green) and negative node pairs (non-edges – colored in red) for homophilic and heterophilic link prediction tasks to contextualize the introduction of these concepts. In these examples, the feature similarity scores of positive node pairs are simplified to follow a uniform distribution, hence we show the horizontal green lines in the Probability Density Function (PDF) plots of the similarity score distribution. Of course in real-world datasets the similarity score distributions for both positive and negative node pairs can be more complex: in Figure 5 in the Appendix we show the *actual* feature similarity score distributions for real-world datasets used in our experiments.
>
> **Q3 Generality of theoretical results on learning rate**
>
> The goal of this theorem is to show that the predicted link probability is related to the magnitude of the threshold M that separates the positive and negative samples, which highlights the different optimizations needed for homophilic and heterophilic link prediction tasks that have not been studied in prior literature. While generalizing the theorem to higher dimensions is non-trivial, our experiments in Section 6 have strengthened this implication to datasets of significantly higher complexity.
>
> **Q4 Details about the real-world data**
>
> | Dataset          | Training Edges | Validation Edges | Test Edges |
> |------------------|----------------|------------------|------------|
> | ogbl-citation2   | 30,387,995     | 86,956           | 86,956     |
> | ogbl-collab      | 1,179,052      | 60,084           | 46,329     |
> | e-comm           | 238,818        | 34,117           | 68,235     |
>
> The above table shows the details of the split of edges for train, validation, and test on real-world datasets. Note that in standard GNN4LP research, there are usually far more training edges than validation and test edges combined (see Table 2 in [14] in paper references). We follow the standard approach of data splitting. It would be an interesting future direction to consider the impact of data splitting on model performance.
>
> **Q5 Reasons for dividing the buckets**
>
> Thank you for the question. We use the minimum degree of connected nodes to divide the buckets in Fig. 4 since it has been shown in previous works that the presence of low-degree nodes (see [44] in paper references) can increase the complexity of heterophily for node classification tasks and negatively impact GNN performance. By using the minimum degree instead of maximum or average degree, we can group edges that are connecting at least one node with low degree and examine how different approaches perform on these edges. We will also explain our choice in the paper.

---

> > ### Comment · Reviewer_YTzD · 2024-08-11
> >
> > Thank the authors for the response to my concerns.

---

### Official Review · Reviewer_dskj · 2024-07-13

**Soundness:** 3
**Presentation:** 3
**Contribution:** 4
**Rating:** 7
**Confidence:** 3

**Summary:**

The paper analyze the impact of heterophily in node features on link prediction tasks, and the authors present a theoretical framework that highlights the different optimizations needed for the homophilic and heterophilic link prediction tasks.

**Strengths:**

The paper analyze the impact of heterophily in node features on GNN performance, the authors argue that the homophilic and heterophilic link prediction tasks should be defined based on how the distributions of feature similarity scores between connected and unconnected nodes are separated, which is innovative. The paper reveals the fundamental differences in optimizations for homophilic and heterophilic link prediction tasks, as well as insights into effective decoder selection for non-homomorphic link prediction tasks.

**Weaknesses:**

1、Some details in the paper are not very clear, such as How is feature extraction implemented? How to generate synthetic graphs that resemble different types of link prediction tasks by varying the feature similarity between connected nodes?
2、In addition, the hierarchical structure of the paper is not very reasonable, such as the chapter positions of the relevant work in Section 5.

**Questions:**

1. How to generate synthetic graphs that resemble different types of link prediction tasks by varying the feature similarity between connected nodes?
2. the variation of the positive feature similarity scores affects the rate of change for the link prediction scores, the authors present definitions with intuitive examples in §3.1 and theoretical analysis in §3.2. but how the predicted link probability is derived?

**Limitations:**

Yes, the author has a clear understanding and explanation of the limitations of the proposed method.

---

> ### Author Rebuttal · Authors · 2024-08-07
>
> **W1, Q1 Clarifications (1) implementation of feature extraction (2) generation of synthetic graphs**
>
> Thank you for your questions.
> - For **W1(1)** implementation of feature extraction, we interpret the "feature extraction" part in your comment as how we obtain the node features for our synthetic and real-world datasets---please correct us if our interpretation is not right. For real-world graphs, we use the original node features provided in the datasets that are created by previous works: ogbl-collab and ogbl-citation2 from Open-Graph Benchmark [1], and e-comm from [2]. We give a brief overview of how the node features are created in Lines 532-539 in Appendix A. Our synthetic datasets are created by rewiring the edges among 10,000 sampled nodes from the real-world dataset `ogbl-collab`; we keep the features attached to each node during the sampling process, which become the node features for our synthetic graphs instead of generating new node features ourselves.
> - For **W1(2) and Q1** generation of synthetic graphs, we describe our synthetic graph generation process in Lines 512-520 in Appendix A (as we mentioned in Line 280 in Section 6.1): the synthetic graphs are generated by randomly sampling 10,000 nodes with their features in ogbl-collab and connecting 2% of all possible node pairs whose feature similarity falls within specified ranges; all graphs share the same set of nodes and features and only differ in their edges. More specifically, we calculate the pairwise feature similarity between all node pairs and create 50-quantiles of feature similarity scores. We select the 3 smallest quantiles, the 3 largest quantiles, and 4 quantiles in equal intervals in between, resulting in 10 quantiles. We then create 10 synthetic graphs by connecting node pairs whose feature similarity scores fall within the same quantile. Thus, by gradually increasing the range of similarity for connected nodes, we create graphs which resemble different types of link prediction tasks and average feature similarity.
>
> [1] Open Graph Benchmark: Datasets for Machine Learning on Graphs. (NeurIPS 2020)
>
> [2] Pitfalls in Link Prediction with Graph Neural Networks: Understanding the Impact of Target-link Inclusion & Better Practices. (WSDM 2024)
>
> **W2 Structure of paper; position of the related work section**
>
> Thank you for your suggestion. We will revise the paper structure and specifically move the related work section to an earlier location (e.g., Section 2) in the final version.
>
>
> **Q2 How is the predicted link probability derived**
>
> Thank you for your question. For our theoretical analysis in Sections 3 & 4, the "predicted link probability" $\hat{y}_{u,v}$ is derived based on the equation in Line 144 ("Theoretical Assumptions" paragraph) in Section 3.2.
>
> We note that we should have used "predicted link score" instead of "predicted link probability" for the discussions in Section 3, as the "link probability" term implies that our prediction $\hat{y}_{u,v}$ is bounded within $[0,1]$ which is not the case in our analysis. However, our analysis can still be given a probabilistic perspective as we can derive the predicted link probability by taking sigmoid $\mathrm{sigmoid}(\hat{y})$ on the predicted link score. We will revise the usage of "predicted link probability" term in our final version.

---

### Official Review · Reviewer_kFpa · 2024-07-15

**Soundness:** 3
**Presentation:** 3
**Contribution:** 3
**Rating:** 6
**Confidence:** 4

**Summary:**

The paper examines how heterophily in node features affects the performance of Graph Neural Networks (GNNs) in link prediction tasks, which typically do not utilize node class labels. It introduces formal definitions of homophilic and non-homophilic link predictions, proposes GNN designs optimized for feature heterophily, and demonstrates through synthetic and real-world data that appropriate decoders and the separation of ego- and neighbor-embeddings can significantly enhance performance in non-homophilic settings.

**Strengths:**

originality: good
quality: good
clarity: good
significance: good

**Weaknesses:**

The theoretical analysis is a bit over-simplified.

**Questions:**

1. It's better to denote the mean feature vector of all nodes as $\hat{x}$ instead of $\hat{X}$

2. Have you tried other heterophily-specific techniques, e.g. high-pass filter or negative message passing?

**Limitations:**

the authors adequately addressed the limitations

---

> ### Author Rebuttal · Authors · 2024-08-07
>
> **W1 Theoretical Analysis**
>
> Thank you for your feedback. First, our theorems are derived under reasonable simplifications, which are typical in the heterophily literature (e.g., [1-2]). Second, our empirical analysis extends well beyond these theoretical assumptions, demonstrating that our theory holds more broadly. Finally, we emphasize that our theoretical analysis aims to provide concise characterizations as an initial step in formalizing this problem. We believe that extending beyond our current theoretical framework will be a valuable direction for future research.
>
> [1] Beyond Homophily in Graph Neural Networks: Current Limitations and Effective Designs. (NeurIPS 2020).
>
> [2] Revisiting Heterophily For Graph Neural Networks. (NeurIPS 2022).
>
>
> **Q1 Notation of mean feature vector: use $\bar{\mathbf{x}}$ instead of $\bar{\mathbf{X}}$**
>
> Thank you for reading our paper carefully! We will make this modification in our final version.
>
> **Q2 Other Heterophily-specific techniques, e.g. high-pass filter**
>
> Thank you for your suggestion. We agree that considering different heterophily-specific techniques is important and will be an interesting direction for future work. The main contribution of this paper is to propose and characterize feature heterophily, marking a pioneering effort in this area.

---

> > ### Comment · Reviewer_kFpa · 2024-08-13
> >
> > Thanks for the reply and I will keep my rating.

---

### Official Review · Reviewer_JA9K · 2024-07-16

**Soundness:** 2
**Presentation:** 4
**Contribution:** 2
**Rating:** 4
**Confidence:** 4

**Summary:**

This paper analyzes the impact of node features in the link prediction task. Based on the feature similarity, it first categorizes and defines link prediction into homophilic, heterophilic and gated ones then shows the differences among them. Further, it explores the encoder and decoder choices along with the ego-neighbor separation for non-homophilic link prediction by theoretical and empirical analysis.

**Strengths:**

* This paper provides a basics including definitions and preliminary conclusion for future non-homophilic link prediction works.
* The overall writing is logical and easy to read.

**Weaknesses:**

* The core conclusions are not exciting enough. For example, the different optimizations needed for homophilic and heterophilic link prediction tasks and the importance of adapting ego-neighbor separation for link prediction are actually intuitive.
* The novelty is insufficient. The theoretical contributions are limited while the solutions are combinations of off-the-shelf modules.
* Only one heterophilic real-world dataset is not enough, which may introduce bias and affect the generality of conclusions.

**Questions:**

Some potential suggestions:
* Add some heterophilic GNNs as the encoders for supplement.
* Design new decoder specifically for heterophilic link prediction tasks.
* Analyze the reasons for the appearance of heterophilic links.

**Limitations:**

* Add more heterophilic real-world datasets to demonstrate the generality of conclusions.

---

> ### Author Rebuttal · Authors · 2024-08-07
>
> **W1&2 Intuitive conclusions & novelty**
>
> We thank the reviewer for the feedback. We note that this is the first work that explicitly characterizes feature heterophily in link prediction with GNNs. Our work pioneers in formalizing the problem, providing concise characterizations, and examining effective designs. Our paper highlights the necessity of careful selection of link prediction designs, including the significant limitation of DOT product decoders, which are widely adopted in the industry, for non-homophilic link prediction tasks, and identifies DistMult as a scalable alternative for settings which offers high scalability. We believe all these are significant and meaningful contributions to the field and open up interesting future avenues for research.
>
> **W3 Heterophilic real-world datasets**
>
> Thank you for your questions. First, we would like to emphasize that we have run experiments on a series of synthetic graphs of different heterophily ratios/feature similarity distributions. Second, going beyond the global characterization of the graphs for the link prediction task, we also provided a zoom-in analysis on the performance on heterophilous edges per method for each real-world dataset, which also demonstrates the effectiveness of our identified designs in improving GNN performance on the full spectrum of low-to-high feature similarity (c.f. Line 360-365, 370-374, and Figure 4). This is important since in reality every dataset exhibits variance in feature similarity across all the links. Finally, by casting light on the problem of link prediction beyond homophily, we hope that our work will motivate the introduction of a high-quality set of benchmarks with diverse feature similarity, beyond the real data that we presented. This trend was also observed in the literature after the challenges of heterophily for GNNs were pointed out in the node classification task [1,2]: as the problem attracted more interest, various research teams dedicated their efforts to introducing a range of heterophilous datasets, thereby advancing findings and methodologies within the subfield. In Figure 1 of our general response plots, we have shown that many real-world heterophilous node classification benchmarks, such as pokec [1] and Amazon ratings [2], are homophilous for link prediction, with most edges connecting nodes with similar features. Nevertheless, as we mentioned above, these datasets still have heterophilous edges where our theory and findings apply.
>
> [1] Large Scale Learning on Non-Homophilous Graphs: New Benchmarks and Strong Simple Methods. (NeurIPS 2021).
>
> [2] A critical look at the evaluation of GNNs under heterophily: Are we really making progress? (ICLR 2022).
>
> **Q: potential suggestions**
>
> Thank you for your valuable suggestions. Following your recommendation, we have added one more heterophilic GNN as the encoder for experiments (highlighted in Table 1 general response pdf) and our conclusions about the GNN decoder and encoder still hold.

---

> > ### Comment · Reviewer_JA9K · 2024-08-13
> >
> > I appreciate the responses from authors.  However, there are still some concerns remaining:
> >
> > * I agree broadly with the author's description of the paper's contributions, but they just don't seem exciting to me. An innovative and feasible solution based on these new formalizations and characteristics is the key for people to be convinced and follow this work.
> > * Synthetic datasets struggle to simulate all the factors that affect performance in real-world datasets (e.g., degree distribution, noise, etc.), which is beyond feature similarity. Therefore, insufficient results on real-world datasets may cast doubt on the validity of the theory and findings in real-world application scenarios.
> >
> > Given the above concerns, I will keep my original rating.

---

> ### Author Response · Authors · 2024-08-13
> **Additional Rebuttal: Results on additional heterophilic dataset**
>
> Following your and reviewer ufZT, YTzD's insightful suggestions, we have identified a range of real-world datasets that exhibit strong feature heterophily (please refer to our [lateset response to reviewer ufZT](https://openreview.net/forum?id=3LZHatxUa9&noteId=2UskaksmAp)). In addition, We have additionally conducted experiments on one heterophilic dataset we identified (Amazon-Computers) and present the result in the table below. We find that the conclusions from our main paper still hold (SAGE > GCN, due to the sparation of the ego- and neighbor-embeddings, and the use of DistMult/MLP decoders is better than DOT).
>
> | Encoder | Decoder  | MRR   | std  |
> | ------- | -------- | ----- | ---- |
> | GCN     | DOT      | 24.67 | 0.61 |
> | GCN     | DistMult | 31.59 | 1.04 |
> | GCN     | MLP      | 52.90 | 0.86 |
> | SAGE    | DOT      | 42.26 | 0.54 |
> | SAGE    | DistMult | 58.42 | 0.12 |
> | SAGE    | MLP      | 58.38 | 0.07 |
> | (NoGNN) | MLP      | 21.15 | 0.23 |
>
> We will include these new results in the final version of the paper.

---

### Author Rebuttal · Authors · 2024-08-07

We would like to express our gratitude to the reviewers for their thoughtful and constructive feedback. We are pleased that all reviewers find the paper clear, most reviewers recognize the novelty of studying the impact of feature heterophily on link prediction with GNNs, and some reviewers (e.g., YTzD, ufZT) point out that they value our theoretical analysis and find that our empirical analyses are comprehensive and validate our theoretical findings.

We answer each reviewer’s questions in separate rebuttals. In this general response, we provide supplementary figures, to which we refer in the individual reviewer questions, as needed.

---

### Comment · Area_Chair_f8zT · 2024-08-11
**Reminder to Check Authors' Reply**

Dear Reviewers,

Would you mind checking the authors' responses?

AC

---

### Decision · Program_Chairs · 2024-09-25

**Decision:**

Accept (poster)

**Comment:**

This paper presents a pioneering approach that explicitly characterizes feature heterophily in link prediction using Graph Neural Networks (GNNs). It highlights the fundamental differences in optimization strategies for homophilic and heterophilic link prediction tasks and provides insights into effective decoder selection for non-homophilic scenarios. However, reviewers have raised concerns about the motivation, novelty, experimental design, and theoretical contributions of the work. Although these concerns have not been fully addressed by the authors, two reviewers have strongly recommended accepting the paper. Therefore, acceptation is suggested.